# Extensive diversity in RNA termination and regulation revealed by transcriptome mapping for the Lyme pathogen *Borrelia burgdorferi*

Emily Petroni[1,5], Caroline Esnault[2,5], Daniel Tetreault[1], Ryan K. Dale [2], Gisela Storz [1] & Philip P. Adams [1,3,4] ✉

Transcription termination is an essential and dynamic process that can tune gene expression in response to diverse molecular signals. Yet, the genomic positions, molecular mechanisms, and regulatory consequences of termination have only been studied thoroughly in model bacteria. Here, we use several RNA-seq approaches to map RNA ends for the transcriptome of the spirochete *Borrelia burgdorferi* – the etiological agent of Lyme disease. We identify complex gene arrangements and operons, untranslated regions and small RNAs. We predict intrinsic terminators and experimentally test examples of Rho-dependent transcription termination. Remarkably, 63% of RNA 3′ ends map upstream of or internal to open reading frames (ORFs), including genes involved in the unique infectious cycle of *B. burgdorferi*. We suggest these RNAs result from premature termination, processing and regulatory events such as *cis*-acting regulation. Furthermore, the polyamine spermidine globally influences the generation of truncated mRNAs. Collectively, our findings provide insights into transcription termination and uncover an abundance of potential RNA regulators in *B. burgdorferi*.

All steps in gene expression – transcription, RNA processing, translation, and protein turnover – determine the ability of a cell to survive. Thus, they are highly regulated in all domains of life. While these regulatory mechanisms have been studied extensively in multiple model systems, such as *Escherichia coli* and *Bacillus subtilis*, much less is known about these processes in other important organisms, such as bacterial pathogens. For example, the fundamental aspects of gene expression have not been studied extensively in the spirochete *Borrelia* (*Borreliella*) *burgdorferi*, the etiological agent of Lyme disease – the foremost vector-borne illness in the United States[1]. The bacterium exists in a complex enzootic cycle[2] that requires acquisition and transmission of *B. burgdorferi* between *Ixodes scapularis* ticks and small vertebrates. The survival and infectivity of *B. burgdorferi* depends on recognition and adaptation to these disparate environments with different physical conditions, temperatures, nutrients, and immune responses. Few studies have examined mechanisms of *B. burgdorferi* gene expression[3,4], particularly transcription termination.

In bacteria, termination has generally been thought to occur by one of two mechanisms: intrinsic (also referred to as factor-independent) or Rho-dependent[5]. Intrinsic termination requires transcription

[1]Division of Molecular and Cellular Biology, Eunice Kennedy Shriver National Institute of Child Health and Human Development, Bethesda, MD 20892, USA. [2]Bioinformatics and Scientific Programming Core, Eunice Kennedy Shriver National Institute of Child Health and Human Development, Bethesda, MD 20892, USA. [3]Postdoctoral Research Associate Program, National Institute of General Medical Sciences, National Institutes of Health, Bethesda, MD 20892, USA. [4]Independent Research Scholar Program, Intramural Research Program, National Institutes of Health, Bethesda, MD 20892, USA. [5]These authors contributed equally: Emily Petroni, Caroline Esnault. ✉e-mail: philip.adams@nih.gov

of a GC-rich RNA hairpin followed by a U-rich tract, inducing RNA polymerase (RNAP) pausing and subsequent dissociation of the transcription elongation complex. Rho-dependent termination requires loading of the hexameric Rho protein complex onto nascent, untranslated RNA at Rho utilization (Rut) sites, which are proposed to be C-rich, G-poor, and unstructured[6]. Rho promotes termination and dislodges RNA polymerase usually 100–200 nt downstream of the Rut site, resulting in RNA 3′ ends that are commonly processed by 3′-to-5′ exonucleases[7,8]. Rho is essential in some bacteria, such as in *E. coli*[9], but not in other species, like *B. subtilis*[10]. Recent studies have challenged the general view of two distinct termination processes. For example, intrinsic and Rho-mediated termination has been observed simultaneously at the same site, which may indicate they can function cooperatively[11–13]. Protein factors, like the transcription-elongation factors NusA and NusG, may also play a role in stimulating intrinsic termination, as recently implicated in *B. subtilis*[13,14].

Studies of intrinsic and Rho-dependent termination in *B. burgdorferi* are limited. Rho has been annotated based on homology with *E. coli* Rho[15], but has not been studied experimentally. Additionally, few studies have sought to determine *B. burgdorferi* intrinsic termination sites. In silico analysis of total RNA-seq data predicted 201 intrinsic terminators[16] and a termination hairpin was identified within the *bmpAB* transcript[17]. However, it remains unknown how the structures of *B. burgdorferi* intrinsic terminators compare to those of other bacteria, what other factors are involved, and whether there is any overlap between termination processes.

In addition to being essential for gene expression, termination can be exploited for gene regulation. Many types of *cis*-acting RNA regulators are associated with secondary structure changes and premature termination and/or RNA processing within 5′ untranslated regions (UTRs) of mRNAs[12,18]. The *cis*-acting RNA structures have been shown to change in response to temperature (RNA thermometers), translation of small upstream open reading frames (uORFs), or by binding *trans*-acting factors, such as metabolites (riboswitches), tRNAs, RNA-binding proteins, or small base-pairing RNAs (sRNAs)[19–24]. sRNAs typically base pair with another RNA to elicit a regulatory response and can be transcribed independently or processed from the 5′ or 3′ UTR of mRNA transcripts, tRNAs, or internal to coding sequences[25]. Therefore, sRNAs can be generated and regulated by premature termination of a longer RNA. In turn, sRNAs themselves have been shown to regulate the termination of RNA targets by modulating Rho accessibility[26]. *B. burgdorferi* RNA-mediated regulation has not been well characterized. Two putative base-pairing sRNAs have been suggested to have regulatory roles and/or impact infectivity[27,28], but their direct base-pairing function has not been demonstrated. Further, no *cis*-acting RNAs have been described in *B. burgdorferi*.

Regulation of *B. burgdorferi* RNA expression, both in culture and during infection, is not well understood. Previous studies examined *B. burgdorferi* gene expression in simulated murine and tick environments[29–31]. However, these studies utilized microarray analysis based on previously annotated open reading frames (ORFs) and did not capture regulatory features like intergenic transcripts or prematurely terminated transcripts. More recently, RNA-seq approaches using spirochetes grown in Barbour-Stoenner-Kelly (BSKII) culture medium[32] have led to further refinement of gene annotations and discovery of potential regulators. One such study estimated RNA boundaries and detected 353 possible sRNAs across bacteriological growth[16]. Another predicted 1,005 putative sRNAs, some dependent on temperature[33]. These studies are limited, however, by their use of total RNA-seq and size-selection RNA-seq respectively, which do not allow for precise resolution of transcript ends.

Multiple RNA-seq approaches have been developed to globally identify both RNA 5′ and 3′ ends[18,34–36]. 5′RNA-seq was previously applied to *B. burgdorferi* and used to discover promoters active during growth in culture and murine infection[37]. Here, we combined the previous 5′RNA-seq data with total- and 3′RNA-seq analysis to systematically update gene annotations, investigate transcription termination, and identify novel regulatory features. We predicted intrinsic terminators using canonical models of RNA hairpin formation/nucleotide composition and experimentally mapped instances of Rho-dependent transcription termination. We documented hundreds of sRNAs and mapped numerous 3′ ends within mRNA 5′ UTRs and internal to coding sequences, indicating that *B. burgdorferi* may harbor numerous RNA regulators. Finally, we demonstrated that the polyamine spermidine impacts the levels of internal 3′ ends and the generation of truncated transcripts. We characterized an RNA fragment that is internal to the spermidine transporter *potB* and is present across the enzootic cycle, particularly in fed ticks. We demonstrated that high extracellular spermidine concentrations led to an accumulation of *potB* 5′ fragments with a concomitant decrease in levels of the full-length mRNA. Taken together, our findings illustrate the power of complementary sequencing approaches to map RNA boundaries, provide insights into fundamental biological processes, and identify new RNA regulators in bacterial pathogens.

## Results

### Global identification of 3′ ends

Initial computational predictions of open reading frames (ORFs) in *B. burgdorferi*[38] overlooked elements such as mRNA boundaries, sRNA genes, and translation initiation from non-canonical start codons. *B. burgdorferi* transcription start sites (TSSs) and 5′ processed RNA ends were previously detected using 5′RNA-seq[37]. Here we sought to identify RNA 3′ ends and combine datasets to globally predict RNA boundaries and refine the *B. burgdorferi* transcriptome annotation.

Three independent cultures of wild-type *B. burgdorferi* B31 (WT) were grown in BSKII to mid-logarithmic phase at 35 °C and then temperature-shifted (~23 °C to 35 °C) to stationary phase (see Methods for details). The temperature shifting (TS) protocol mimics the tick (23 °C) and mammalian (37 °C) temperatures and effectively induces *rpoS*[39,40], the global stress-induced alternative sigma factor active during mammalian infection. Total RNA was isolated from the logarithmic (log) and TS-stationary phase cultures and analyzed using modified RNAtag-seq (total RNA-seq)[41] and 3′RNA-seq (also called Term-seq)[12,18] (Figs. 1 and S1). The replicate total RNA-seq and 3′RNA-seq datasets were highly correlated (Fig. S2A).

We curated a list of statistically significant RNA 3′ ends (Supplementary Data 1) and cataloged them based on their distance to ORFs and other genomic features (see Methods for details). The total numbers of identified 3′ ends were 1,333 and 944 for cells grown to logarithmic or TS-stationary phase, respectively. The *B. burgdorferi* strain is the same used previously for the 5′RNA-seq conducted on logarithmic grown spirochetes[37]. Therefore, we integrated these three RNA-seq datasets (5′RNA-seq, total RNA-seq, and 3′RNA-seq) for the exponentially growing cells.

### Mapping mRNA boundaries in B. burgdorferi

Most RNA boundaries in *B. burgdorferi* have not been experimentally mapped, even for the well-studied infection-essential genes *ospAB*, *ospC* and *pncA* (Fig. 2). We first examined the *ospA/B* (*bba15/16*) operon, which encodes the outer surface protein (OspA) required for *B. burgdorferi* tick colonization[42]. The combination of 5′RNA-seq[37], total RNA-seq, and 3′RNA-seq allowed us to identify a TSS and multiple 3′ ends throughout the *ospA/B* operon (Fig. 2A). Northern analysis using a probe upstream of the dominant downstream 3′ end (blue asterisk) documented a polycistronic *ospAB* transcript consistent with co-transcription of the genes. A previously-predicted[33], 3′ derived sRNA (SR0885) was also detected. Additional northern analysis with oligonucleotide probes upstream of two other 3′ ends in this operon revealed a novel 5′ derived sRNA and a previously-predicted ORF-internal sRNA (SR0884)[33] (Fig. S2B). Other fragments of *ospA* or *ospB*

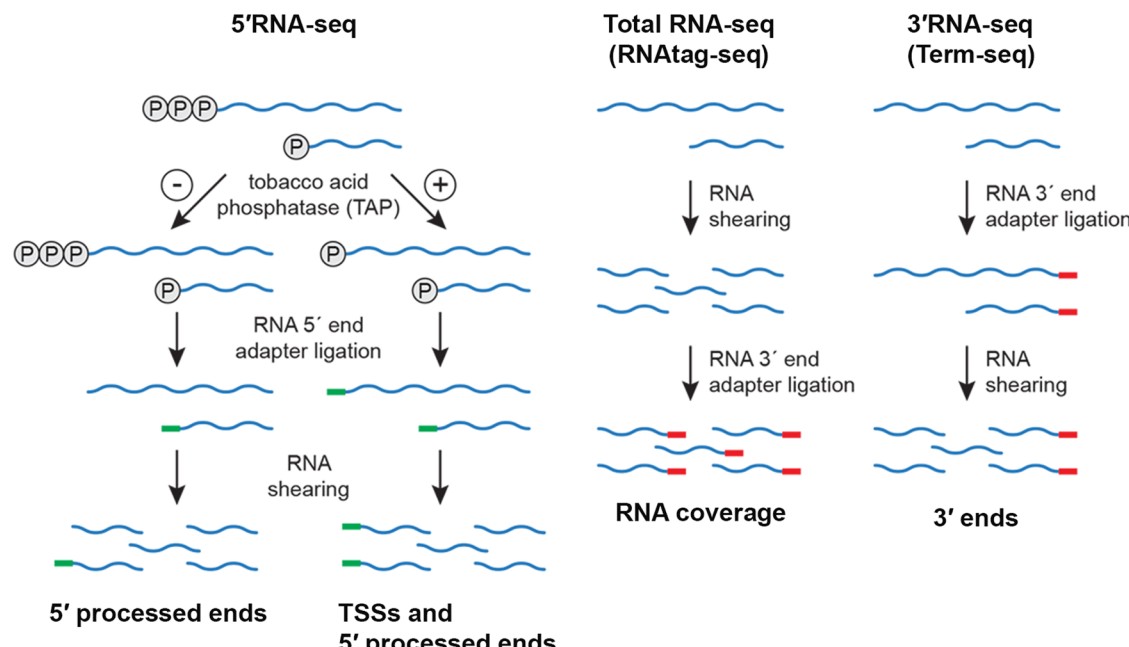

**Fig. 1 | RNA-seq approaches.** Schematic of adapter ligation and RNA shearing for 5′ RNA-seq[37], total RNA-seq (modified from the RNAtag-seq methodology[41]) and 3′ RNA-seq (modified from the Term-seq methodology[18]). 5′ phosphates (circled 'P'), RNA 5′ end adapter (green line) and RNA 3′ end adapter (red line) are indicated. 5′ RNA-seq identifies and distinguishes transcription start sites (TSSs) from processed 5′ ends. Total RNA-seq captures sequencing read coverage across transcripts. 3′ RNA-seq identifies RNA 3′ ends. Expanded figure in Supplementary Information (Fig. S1); modified from ref. 12.

transcripts, which are likely processed from the full-length mRNA, were also detected using these probes (Fig. S2B). This analysis reveals the complexity that can exist within just a two-ORF operon.

We observed that some genes harbor long 3′ UTRs (Supplementary Data 1). For example, 3′RNA-seq suggested that the *ospC* mRNA, encoding a well-studied outer surface protein required for *B. burgdorferi* mammalian colonization[43], harbors a 3′ UTR of 91 nt (Fig. 2B). Northern analysis with a probe directly upstream of the identified 3′ end (blue asterisk) confirmed the expected ~744 nt size of the *ospC* mRNA, including the predicted 3′ UTR. Higher levels of *ospC*, a prototypical member of the *rpoS* regulon[44], were observed in the TS-stationary RNA-seq data and by northern analysis, consistent with RpoS induction by the TS protocol. Further, the *ospC* 3′ UTR encodes novel 3′-derived sRNAs of ~80 and ~100 nt in length, also present at higher levels in the TS-stationary sample.

We additionally noted genes that harbored long 5′ UTRs with premature 3′ ends (before the end of the coding sequence). The *pncA* gene, encoding a nicotinamidase essential for infection[45], had a 3′ end directly upstream the start of the coding sequence (Fig. 2C). A previously predicted[33] sRNA (SR0754) also had this 3′ end and total RNA-seq reads suggested that *pncA* has a 290 nt 5′ UTR from which SR0754 is derived. Northern analysis with a probe directly upstream of the identified 3′ end (blue asterisk) confirmed co-transcription of SR0754 and *pncA*. Collectively, these data revealed complex transcriptional units, long UTRs, and multiple 3′ ends. These examples also illustrate how 3′ ends could be reflective of regulatory events and/or be associated with the generation of regulatory sRNAs.

**Mapping sRNA boundaries in B. burgdorferi**

Two previous *B. burgdorferi* studies predicted either 1005 sRNAs by size-selection RNA-seq[33], or 353 sRNAs by total RNA-seq[16]. We surveyed the 5′ and 3′ ends for these sRNAs in our dataset for logarithmically-grown *B. burgdorferi* (see Methods for details). This resulted in a list of 317 detectable sRNAs (Supplementary Data 2). We annotated possible TSS(s), 5′ processed end(s), and abundant 5′ and 3′ ends for each sRNA. In nearly all cases, the sRNA boundaries were modified from the

previously reported boundaries, based on the 5′RNA-seq and 3′RNA-seq data. For several loci there were significant differences. One example is SR0947. Popitsch et al., predicted two sRNAs in this region, and Arnold et al., predicted one, though neither aligned with 5′ and 3′ mapping (Fig. 2D). Northern analysis using a probe directly upstream the identified 3′ end (blue asterisk), documented a ~143 nt RNA predicted by our mapping.

In some cases, we observed multiple forms of an sRNA, as has been reported for other sRNAs across bacteria[25]. For example, two TSSs and one 3′ end were detected for SR0961 (Fig. 2E), which would result in two forms of the sRNA at ~147 nt and ~78 nt. The expected sRNA products were verified by northern analysis using a probe directly upstream of the sRNA 3′ end (blue asterisk) (Fig. 2E). An additional probe upstream from the second TSS only detected the larger sRNA product (Fig. S2C). This was missed in previous annotations of SR0961. In one study SR0961 was assumed to be intergenic with two 3′ ends and a possible small ORF (sORF)[33], but the predicted boundaries of the sORF do not fall within the sRNA we detect. SR0961 shares a 3′ end with *bbb13*, albeit the levels of SR0961 are much higher than *bbb13* (Fig. S2D). SR0947 and SR0961 are representative examples where precise RNA-seq transcript mapping refines previous sRNA annotations. Accurately understanding sRNA sizes and their various forms is important for future functional characterization of these RNAs.

**Improving B. burgdorferi transcriptome annotation**

Using the list of sRNAs (Supplementary Data 2), current NCBI and UniProt gene annotations for mRNA ORFs, tRNAs and rRNAs, TSSs for mRNAs[37] and 3′ ends for mRNAs (Supplementary Data 1), we assembled a transcriptomic annotation file (Supplementary Data 3; see Methods for details). This file lists our best predictions for the boundaries of ORFs, 5′ UTRs, 3′ UTRs, rRNAs, tRNAs, and sRNAs for *B. burgdorferi* grown to logarithmic phase. In many cases, these predictions indicate complex gene arrangements. For example, the annotations for the *bb0404* and *bb0405* genes suggest both genes could be independently transcribed, with the *bb0405* 5′ UTR overlapping *bb0404* and

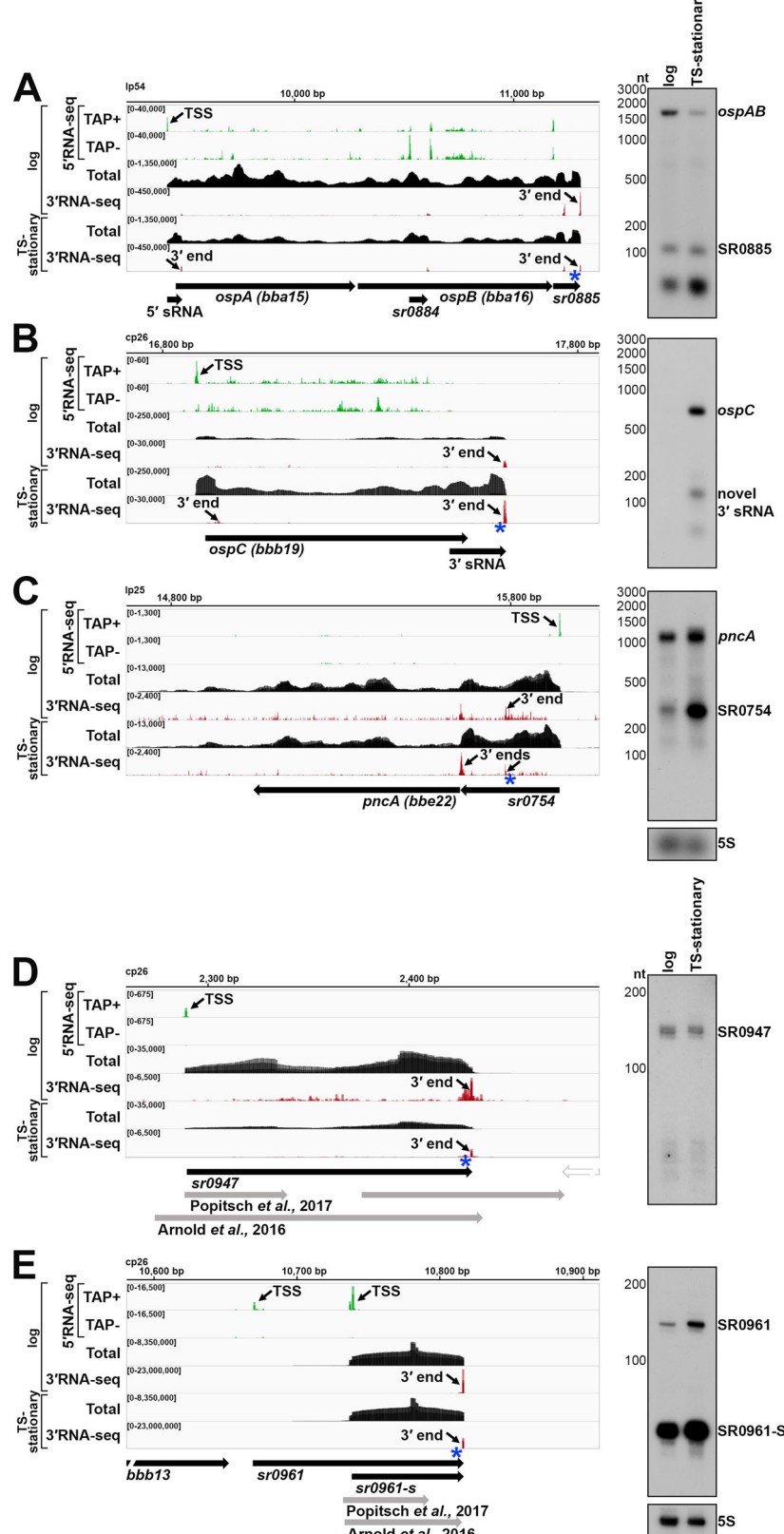

encoding an sRNA (SR0297) (Fig. S3A). *bb0404* also harbors a novel 5′ derived sRNA. For most cases, the predictions match manual inspection of the RNA-seq data, though in a few cases the predictions will require additional experiments to make conclusive determinations of RNA boundaries. For example, there are numerous TSSs and 3′ ends detected within/surrounding genes *bb0397* and *bb0398* (Fig. S3B),

making delineation of the transcriptional units more challenging. For some sRNAs, an abundance of TSSs or 3′ ends made it difficult to predict the ends (Fig. S3C and S3D). Collectively, these annotations are predictions for the *B. burgdorferi* transcriptome during logarithmic growth and should be continually refined with future studies, including other growth conditions. Understanding gene boundaries is critical

**Fig. 2 | 5′-, total-, and 3′RNA-seq revise transcript boundaries and reveal novel UTR-derived RNAs and sRNAs.** RNA-seq browser image (left) and northern analysis (right) for the (**A**) *ospAB* locus, (**B**) *ospC* locus, (**C**) *pncA* locus, (**D**) SR0947 sRNA and (**E**) SR0961 sRNA are shown. Browser images display sequencing reads for one DNA strand for RNA isolated from logarithmic and temperature-shifted (TS) stationary phase cells. 5′RNA-seq tracks (green) represent an overlay of two biological replicates[37], while total- (black) and 3′RNA-seq (red) tracks represent an overlay of three biological replicates. Read count ranges are shown in the top left of each frame. The chromosomal coordinates, the relative orientation of ORFs and small RNAs (wide black arrows), previous sRNA annotations[16,33] (wide gray arrows), 3′ ends from Supplementary Data 1 (small black arrows), and select TSSs (small black arrows) as determined by the ratio of reads between ±TAP tracks[37], are indicated.

Genes annotated on the opposite DNA strand are indicated by a wide white arrow. For northern analysis, cells (PA003) were grown to a density of $5.5 \times 10^7$ cells/ml (log) or $3.5 \times 10^8$ cells/ml following temperature shifting (TS-stationary) after dilution of the starter culture and lysed. Total RNA was extracted, separated on an agarose (**A**–**C**) or polyacrylamide (**D**, **E**) gel, transferred to a membrane, and probed for the indicated RNAs. Sample and molecular weight marker (nt) labels are the same for **A**–**C** and **D**, **E**; RNAs were probed sequentially on the same membrane. The approximate probe sequence location is indicated by the blue asterisk on the corresponding browser image. The membranes were also probed for 5S as a loading control. A short unannotated RNA was detected in the *ospAB* 3′ UTR, likely a shorter fragment of SR0885. A previously unannotated fragment in the *ospC* 3′ UTR was denoted as 'novel 3′ sRNA'.

for designing and analyzing genetic manipulations and interpreting differential gene expression data.

### Features of B. burgdorferi 3′ ends

We focused on global analyses of RNA 3′ ends to identify molecular signatures of termination and regulatory elements. Detected 3′ ends were first subclassified (see Methods for details) according to their locations relative to annotated genes (Fig. S4A). 3′ ends that could not be assigned to one unique category were counted in multiple categories. For logarithmic grown cells (Fig. S4B), 26.4% (444) of 3′ ends mapped <100 bp downstream of an annotated gene (primary 3′ ends), likely attributable to transcription termination. These primary 3′ ends mapped downstream of 152 mRNAs, 3 rRNAs, 20 tRNAs, and 285 sRNAs (16 mRNAs were predicted to share a 3′ end with an sRNA). The majority, (59.1%; 994) of 3′ ends mapped internal to ORFs (internal 3′ ends), and 3.9% (65) were classified as orphan. Many in this orphan category were upstream of mRNAs. These latter examples could be instances of premature transcription termination or RNase 3′ processing, a hallmark of RNA-mediated regulation in *E. coli*[12] and *Bacillus subtilis*[18]. The *B. burgdorferi* TS-stationary 3′ ends had a similar distribution (Fig. S4B). A subset (33.8%) were uniquely detected in the TS-stationary phase samples (Fig. S4C).

3′ UTR length was calculated for mRNA primary 3′ ends (152 total) detected for logarithmic-grown cells using the annotated stop codon (Supplementary Data 1). These data showed 3′ UTRs span a variety of lengths (Fig. S4D) with 74 (48.7%) UTRs of ≥60 nt. The longer 3′ UTRs could correspond to sites of mRNA regulation, and/or encode alternative ORFs or be substrates for sRNA generation. In other bacteria, many 3′ UTR-derived sRNAs act as *trans*-encoded, base-pairing RNA regulators[25].

### Predicted intrinsic terminators

Little is known about transcription termination in *B. burgdorferi*. Therefore, we analyzed the 3′ ends identified by 3′RNA-seq (Supplementary Data 1) together with the surrounding sequences. We first examined the G/C content surrounding logarithmic phase-identified 3′ ends. Typically, higher G/C content is required to form strong 3′ hairpin structures for intrinsic termination and/or is required for the stability of 3′ ends generated by processing or Rho-dependent termination[7,46,47]. We calculated the average G/C content immediately surrounding the detected 3′ ends as well as regions 50–100 nt upstream of 3′ ends, as a control (Fig. 3A). There was slight average enrichment of G/C nucleotides in an ~25 bp region upstream of 3′ ends, with a return to average usage immediately downstream of 3′ ends.

To identify 3′ ends that could correspond to intrinsic terminators, we applied a previously-described quantitative method[48]. This approach is based on the canonical model (using known *E. coli* and synthetic terminator sequences) of intrinsic termination – RNA hairpin formation and a poly-U-tract causing transcriptional pausing to ultimately trigger termination. Sequences 50 nt upstream and 10 nt downstream of each 3′ end were analyzed and given an "intrinsic termination score" (Supplementary Data 1), where a score over 3.0 is

predictive of intrinsic termination. Based on these analyses, we predicted 253 (19.0%) of the 3′ ends detected for cells captured in logarithmic-phase growth had secondary structures with high scores for intrinsic termination (terminator score ≥ 3.0). We observed 44.5% (61/137) of 3′ ends downstream of an ORF had a significant terminator score compared to 15.2% (151/994) ORF-internal 3′ ends with a significant terminator score. In the total 253 predicted intrinsic terminators, the average size of the stem was ~7 nt with a loop of 11 nt (Fig. 3A). On average, the stem loop was preceded by an A-tract of 3 adenines, 2 being consecutive, and followed by a U-tract of 5 uracil nucleotides, 3 being consecutive. MEME motif analysis on the same putative terminator sequences identified a conserved logo for 86 of the 3′ ends. This motif notably included the canonical poly-U enrichment downstream of the 3′ end (Fig. 3A).

One other *B. burgdorferi* study[16] predicted intrinsic terminators using total RNA-seq analysis in combination with prediction programs, such as TransTerm[49]. When we compared our list of putative terminators to those predicted by Arnold et al., within a window of 10 nt, there was negligible (8.4%) overlap (Fig. S4E). Even after increasing the comparison to a window of 100 nt, only a subset (23.8%) of the terminators predicted by Arnold et al., were found in our dataset (Fig. S4E). We also compared our 3′ ends to those computed by TransTerm for ORFs annotated in WT *B. burgdorferi* B31 (strain PA003). 38.7% and 60.1% of our predicted intrinsic terminators were computationally predicted by this software using windows of 10 or 100 nt, respectively (Fig. S4F).

Overall, we found limited examples of *B. burgdorferi* 3′ ends that harbored canonical features of intrinsic termination. The defining principles of intrinsic termination are largely based on the study of model bacterial species, which is also the basis of software termination-prediction tools. Therefore, we sought to directly compare *B. burgdorferi* 3′ ends to the ends in other, well-studied organisms.

### Comparison of predicted intrinsic terminators across bacterial species

Global 3′ end mapping by 3′RNA-seq (Term-seq) has been performed on a variety of bacterial species, including gram-negative *E. coli*[12] and *Pseudomonas aeruginosa*[50] and gram-positive *B. subtilis*[18]. The genomic A/T nucleotide usages of these organisms are 49.2%, 33.4%, and 56.5% respectively, compared to 71.8% for *B. burgdorferi*. We acquired the raw 3′RNA-seq data from these studies, analyzed the reads with the same computational pipeline, and compared the identified 3′ ends.

Across *E. coli* (Fig. 3B), *P. aeruginosa* (Fig. 3C), and B. *subtilis* (Fig. 3D), G/C usage increased in a ~25 nt region upstream of 3′ ends. The increase was significantly more pronounced than what was observed for *B. burgdorferi*, especially for *P. aeruginosa* and *B. subtilis* where two clear G/C content peaks emerged, likely corresponding to the stem sequence of the secondary structure upstream of 3′ ends. We examined the RNA secondary structures surrounding the 3′ ends for each bacterium using the same quantitative method[48]. This predicted 30.0% *E. coli*, 50.1% *P. aeruginosa*, and 73.6% *B. subtilis* of identified 3′ ends as possible intrinsic terminators (terminator score ≥3.0). Average

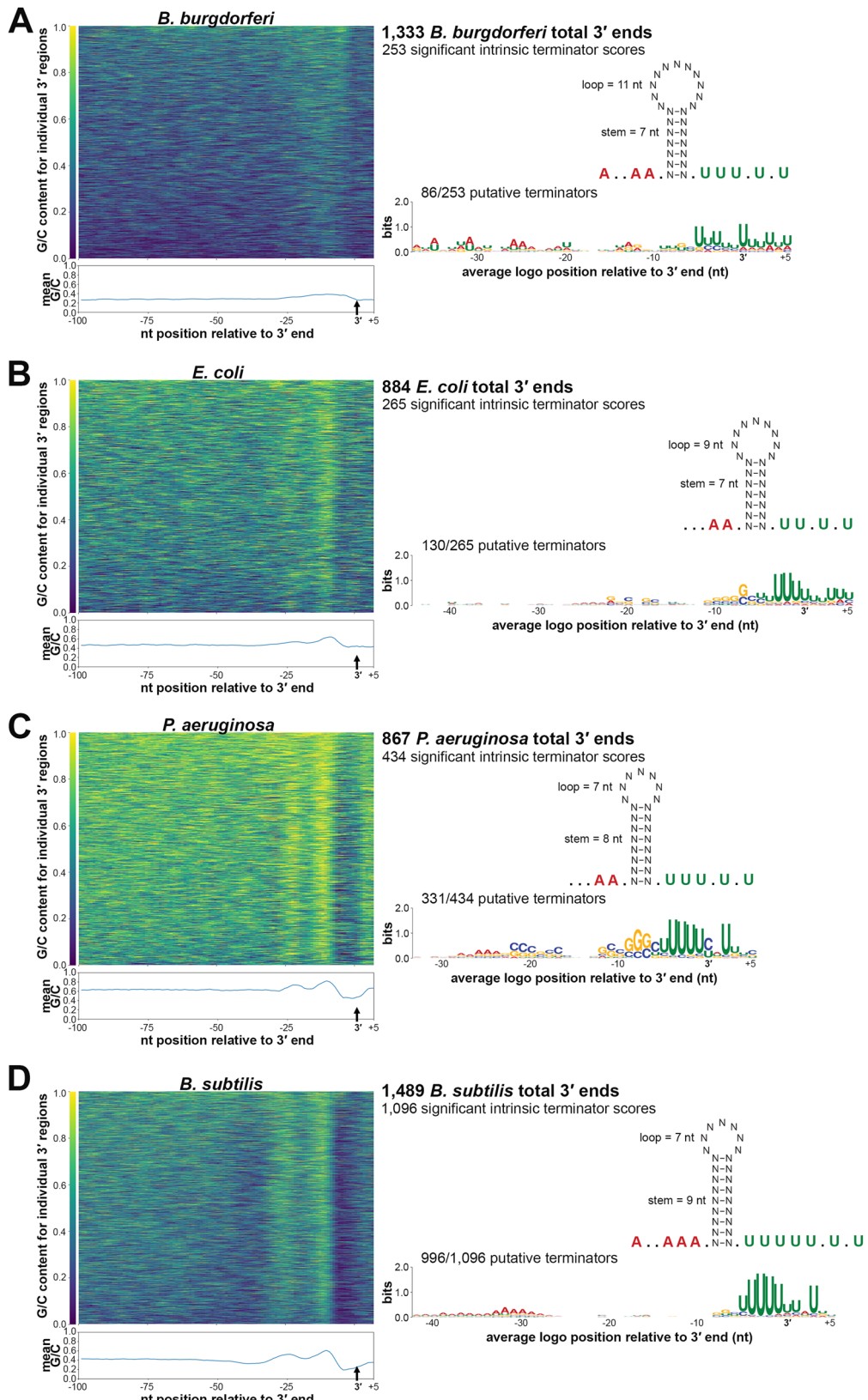

terminator secondary structures for each organism were similar to the average structure for *B. burgdorferi*. The mean stem for *B. subtilis* was the longest at ~9 nt and *B. burgdorferi* had the longest average loop at ~11 nt. *B. subtilis* terminators also harbored the longest stretch of poly-A and -U. MEME analysis of sequences surrounding the 3′ ends with

significant intrinsic terminator scores showed the expected poly-U enrichment for each bacterium. The *P. aeruginosa* termination logo also contained a conserved C/G rich stem sequence, and the *B. subtilis* logo, which was found in 90.9% of predicted terminators, had the strongest conserved A-track. Collectively, these data indicated general

**Fig. 3 | Comparison of B. burgdorferi 3′ ends and predicted terminator structures to those in other bacteria.** G/C content of sequences surrounding 3′ ends (left) and average terminator structure predictions and logos (right) for (**A**) *B. burgdorferi*, (**B**) *E. coli*, (**C**) *P. aeruginosa*, and (**D**) *B. subtilis* are shown. For *B. burgdorferi*, identified 3′ ends from 3′RNA-seq (logarithmic phase samples, Supplementary Data 1) were analyzed. For other bacteria, Term-seq reads from logarithmic phase samples in previous studies[12,18,50] were analyzed by our computational pipeline to identify 3′ ends (see Methods). Total numbers of 3′ ends identified for each organism are given. Average G/C content, represented by a heatmap (left panels) was computed by analyzing sequences -100 to +10 nt using a 5 nt moving window, relative to identified 3′ ends (position 0, represented by a small black vertical arrow). Fraction of G/C content at a position is indicated by the color scale displayed to the left of each heatmap. Mean G/C enrichment for every position is plotted below each heatmap (blue line). An "intrinsic terminator score"[48] and KineFold[111] structure prediction was obtained for each 3′ end. The number of 3′ ends with significant scores (≥3.0) for each organism are listed (right panels). Also shown is the average terminator structure, using all 3′ end sequences with a significant intrinsic terminator score. Sequences upstream and downstream of the terminator hairpin ranged from 6-10 nt as indicated by periods and A residues (red font, representing the average poly-A tract and consecutive adenine nucleotides) and periods and U residues (green font, representing the average poly-U tract and consecutive uracil nucleotides), respectively. The average lengths of the terminator hairpin and stem-loop are also listed in the structure. For terminator logo predictions, MEME analysis discovered motifs (*B. burgdorferi*, E = 4.2 × 10⁻²; *E. coli* E = 1.5 × 10⁻¹²; *P. aeruginosa*, E = 3.4 × 10⁻⁴⁵; *B. subtilis*, E = 2.7 × 10⁻¹¹²) for each organism. Relative enrichment of a given base is indicated by the height of the letter at that position. Fractions correspond to number of sequences containing the motif over the total number analyzed (3′ end sequences with a significant intrinsic terminator score).

---

features of intrinsic termination are present across all bacteria examined. However, there were species-specific differences, with fewer *B. burgdorferi* 3′ ends preceded by canonical intrinsic terminators.

## Mapping of Rho termination regions

The other mechanism of bacterial termination involves Rho, a hexameric protein complex that binds and displaces RNA polymerase near C-rich, G-poor, unstructured Rho utilization (Rut) sites[6]. Rho has been annotated in *B. burgdorferi*[15], but no Rho-terminated genes have been reported. To experimentally map possible instances of Rho-dependent termination we treated cells with bicyclomycin (BCM), the selective inhibitor of Rho. Two independent *B. burgdorferi* cultures grown to logarithmic phase were split, with half left untreated and the other half treated with 100 μg/ml BCM for 6 h prior to RNA isolation. This concentration of BCM had marginal effects to *B. burgdorferi* growth in 6 h, but ultimately resulted in cell death (Fig. S5A). Total RNA was analyzed by total RNA-seq (±BCM-seq). The replicate ±BCM-seq datasets were highly correlated (Fig. S5B). Using these data, we calculated a "Rho score" for each genomic position as done previously for *E. coli*[12] (see Methods for details). This ratio reflects the degree of readthrough when Rho-activity is inhibited, where a score of >2.0 is indicative of at least a twofold increase in readthrough in the +BCM compared to the -BCM sample. We found 393 genes to have regions putatively associated with a Rho termination event – downstream of 265 annotated ORFs, 26 tRNAs, and 102 sRNAs in both biological replicates (Supplementary Data 4). These genomic positions more likely correspond to processed RNA 3′ ends than the termination sites, since Rho-terminated transcripts typically are processed by 3′ to 5′ exonucleases[7,8], and *B. burgdorferi* harbors the conserved PNPase exonuclease[38,51]. Hence, we refer to the identified genomic locations as "Rho termination regions".

Several genes with significant Rho scores were selected for northern analysis. In other bacteria, Rho auto-regulates its own transcription termination[52–54]. In *B. burgdorferi*, where *rho* is co-transcribed in a multi-gene operon, there was significant readthrough of *rho* upon BCM treatment and a Rho termination region was detected downstream of the *rho* stop codon (Fig. 4A). Northern analysis of the RNA collected for ±BCM-seq using a probe to *rho* revealed increased levels of the full-length operon and higher molecular weight bands in the +BCM sample. This strongly suggests that that *B. burgdorferi* Rho also auto-regulates its own termination. Browser images of other loci with Rho termination regions showed similar BCM-dependent readthrough of 3′ ends, including the mRNA *bptA*, encoding a membrane-associated tick infectivity protein[55] (Fig. 4B), and the mRNA *bosR*, encoding an oxidative stress regulator[56] (Fig. 4C). Northern analysis showed increased levels of the mRNAs and longer transcripts with the addition of BCM. For the *bosR* northern blot, many bands were observed in the +BCM lane, which could reflect instability of the full-length transcript when Rho is inhibited.

There were also 245 instances of Rho termination regions within mRNA coding sequences, identified in both ±BCM-seq replicates (Supplementary Data 4). Often these regions were near genes encoding predicted sRNAs. For example, *panF* encodes a putative pantothenate permease[38] and had an internal 3′ end, which also mapped to SR0694 and was directly followed by a Rho termination region (Fig. 4D). Increased levels of both the sRNA and full-length *panF* were detected by northern analysis upon the addition of BCM. Other *panF* transcripts were also observed in the +BCM sample, suggesting these are the result of Rho readthrough. The hypothetical protein *bbk15* on the mammalian infection-essential replicon lp36[57], has a predicted 5′ derived sRNA (SR0821) within its 5′ UTR with a Rho termination region within the *bbk15* ORF (Fig. 4E). Northern analysis showed SR0821 is the most abundant transcriptional product in untreated cells, but SR0821 levels decreased and *bbk15* full-length mRNA levels increased upon BCM treatment.

Finally, many independently transcribed sRNAs were also associated with Rho termination events (102 downstream of and 5 internal to sRNAs). For example, SR0742, an annotated ~54 nt sRNA on replicon lp21, had significant transcriptional readthrough with a predominant longer form of the sRNA in the +BCM sample (Fig. 4F). These observations suggest a Rho-mediated mechanism to generate sRNA gene products, which to our knowledge has not been described for another bacterial sRNA. Collectively, these data identify likely sites of Rho termination that can be at the beginning, middle or end of genes across the spirochete genome.

## Features of Rho termination regions

Given that Rut sites are typically C-rich and G-poor, we calculated C:G nucleotide usage for all significantly predicted Rho termination regions, as well as for a control group of randomly selected genomic coordinates. Relative to the control group, there was a higher local C:G ratio within ~200 nt of the 3′ ends associated with Rho termination regions (Fig. 5A), similar to, though less pronounced than *E. coli* Rho regions identified by ±BCM-seq[12].

We observed 80 (16.2%) Rho-termination events that overlapped with predicted intrinsic terminations (searching within and 800 bp downstream of annotated genes), which is nearly half (45.2%) of all predicted intrinsic termination associated with mRNAs (Fig. 5B). This was also observed in *E. coli* for 31.2% of Rho-regions[12]. Recently, some studies have challenged the notion that Rho-mediated and intrinsic termination are mutually exclusive, suggesting cooperation at some terminators[11,13]. We selected a few genes with significant Rho and intrinsic termination scores for northern analysis. The gene *manA*, encoding the enzyme mannose isomerase[38], had a Rho termination region detected downstream of the stop codon and a primary 3′ end with a high (12.5) intrinsic termination score (Fig. 5C). The *ospD* transcript, encoding an outer surface lipoprotein, which also had a Rho termination region, had a lower (6.4) intrinsic termination score

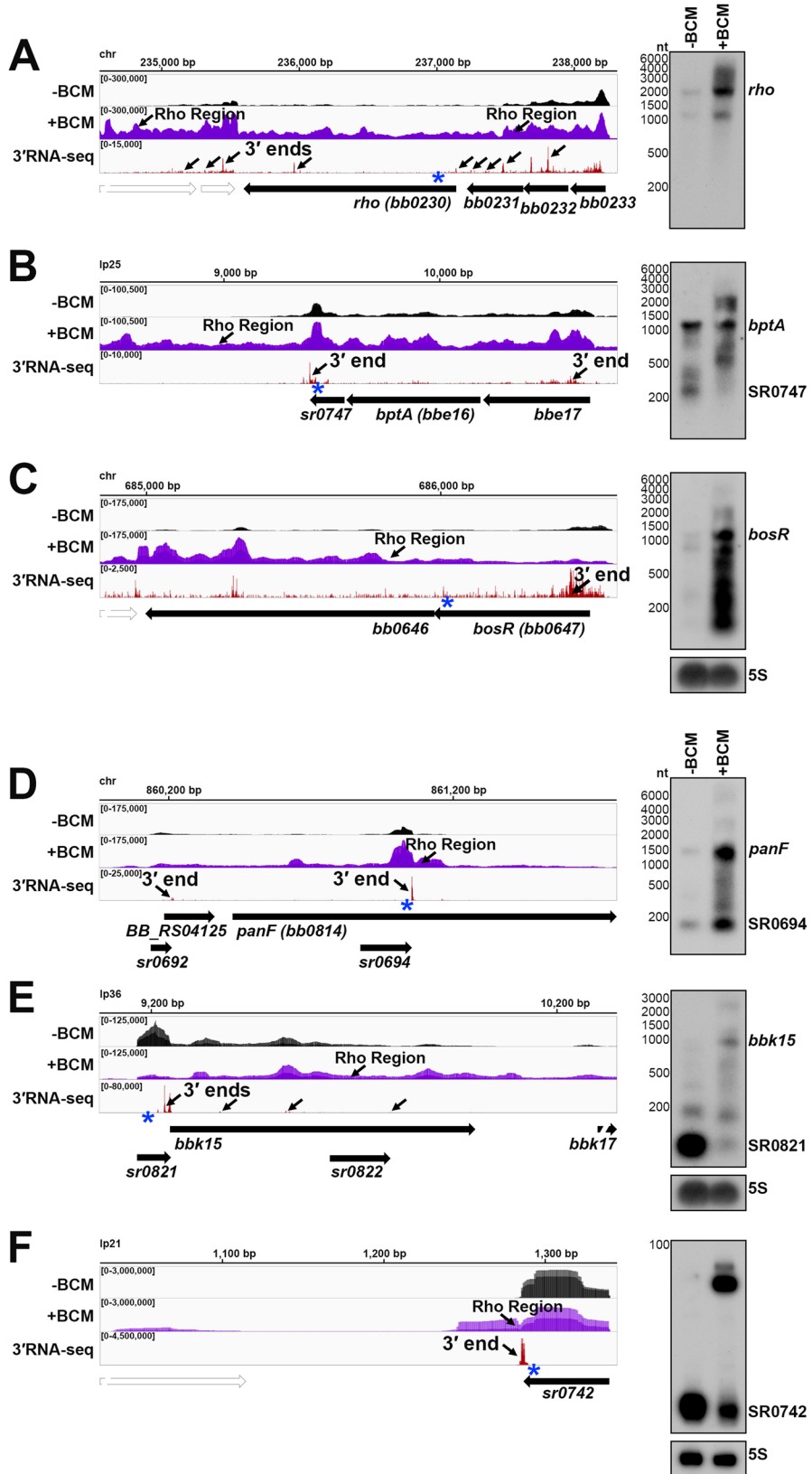

(Fig. 5D). The levels of full-length *manA* slightly increased in the +BCM sample, whereas levels of full-length *ospD*, and a 3′ derived sRNA, SR0852, decreased considerably. Interestingly, for both genes, we detected an accumulation of shorter *manA* and *ospD* fragments with the addition of BCM. These data support a hypothesis that the combination of Rho and intrinsic termination may facilitate efficient termination and increase RNA stability to ensure the generation of full-length transcripts. Additional investigations of these events in *B. burgdorferi* and other bacteria are required to parse out the relative contributions of intrinsic and Rho-dependent termination to any one 3′ end. Termination events and/or 3′ end formation can also be conditional and important for *B. burgdorferi* gene regulation.

**Fig. 4 | Examples of Rho-dependent termination in *B. burgdorferi*.** RNA-seq browser image (left) and northern analysis (right) for identified Rho regions downstream of the (**A**) *rho* mRNA, (**B**) *bptA* mRNA, and (**C**) *bosR* mRNA, or internal to/associated with an sRNA for (**D**) *panF* mRNA, SR0694 (**E**) *bbk15* mRNA, SR0821 and (**F**) SR0742 sRNA are shown. Browser images display data for one DNA strand for ±BCM-seq reads (black and purple), which each represent an overlay of two biological replicates and 3′RNA-seq tracks (red), which represent an overlay of three biological replicates. Read count ranges are shown in the top left of each frame. The chromosomal coordinates, relative orientation of ORFs and small RNAs (wide black arrows), 3′ ends from Supplementary Data 1 (small black arrows) and Rho termination regions (small black arrows; approximated by the average location between both biological replicates) are indicated. Genes annotated on the opposite DNA strand are indicated by a wide white arrow. For northern analysis, cells (PA003) were grown to a density of $4.0 \times 10^7$ cells/ml (log) after dilution of the starter culture, the culture split and half treated with BCM for 6 h. Cells were lysed and total RNA was extracted, separated on an agarose or polyacrylamide gel, transferred to a membrane and probed for the indicated RNAs. Sample and molecular weight marker (nt) labels are the same for **A**–**C** and **D**–**F**; RNAs were probed sequentially on the same membrane for **A**–**E**. The 5S blot in panel **C** was repeated in panel **E**. Approximate probe sequence location is indicated by the blue asterisk on the corresponding browser image. Intrinsic termination scores for the primary and/or dominant 3′ end for each gene (Supplementary Data 1) were determined as: *rho* = 2.5; *bptA* = n/a; *bosR* = n/a; *panF* = 1.99; *bbk15* = 1.32; *sr0742* = 1.48. Additional northern analysis was performed using two independent ±BCM biological replicates and showed the same results for *rho*, *bosR*, and *bptA* RNAs.

## Growth-dependent effects on upstream and internal 3′ ends

A previous study demonstrated that 3′ ends in 5′ regions of mRNAs in *E. coli* could be the result of varied regulation[12] including the binding of small base-pairing RNAs (sRNAs), the generation of 5′ derived sRNAs, translation of an upstream open reading frame (uORF), or conformational changes in *cis*-acting RNA elements (like RNA thermometers and riboswitches). Thus, we collated upstream and ORF-internal 3′ ends that could correspond to such regulatory events for *B. burgdorferi* (see Methods for details), resulting in a list of 870 3′ ends (Supplementary Data 5) for our logarithmic dataset.

We tested for the generation of shorter RNAs for several of the 3′ ends in 5′ mRNA regions by carrying out agarose gel northern analysis for cells across growth with temperature-shifting using a single probe in the 5′ region (blue asterisks, Fig. 6). As a control we probed for *ospC* (Fig. 6F), whose levels increase across growth. A diversity of mRNAs was surveyed, with an emphasis on genes crucial for infection and/or encoding important enzymes and transporters. We expected to detect distinct bands for short 5′ RNAs, if RNA fragments were generated, as well as full-length mRNAs and hypothesized that changes in the abundance or ratio of 5′ to full-length RNA levels could signify regulation.

We previously documented (Fig. 4E) that *bbk15* harbors a 5′ derived Rho-dependent sRNA (SR0821). Northern analysis across growth showed a dominant band corresponding to the expected SR0821 size (~82 nt) whose levels clearly increased in the temperature-shifted sample (Figs. 6A and S6A). The levels of longer full-length *bbk15* transcripts were lower but also increased with time.

*bgp* encodes a dual-function glycosaminoglycan (GAG) binding adhesin and 5′ methylthioadenosine/S-adenosyl homocysteine (MTA/SAH) nucleosidase[58–61]. Bgp is predicted to be important for facilitating spirochete adhesion to the host extracellular matrix and for eliminating MTA/SAH, toxic byproducts resulting from adenine/methionine salvaging. We detected a novel short 5′ RNA (~58 nt) that corresponded to the 5′- and 3′RNA-seq signals (Figs. 6B and S6B). This also overlapped a previously predicted[33] longer sRNA (SR0458) of ~333 nt that may correspond to the entire *bgp* 5′ UTR (195 nt), although the boundaries for this longer sRNA are not clearly defined in our RNA-seq data. The levels of these shorter 5′ RNAs decreased across growth with a concomitant increase in the levels of a transcript encoding *bgp* and the downstream *pta* gene. The difference in the ratio of the 5′ fragments to the full-length *bgp* RNA could be indicative of regulation.

We next probed four genes whose products are part of transport systems, as such systems may be regulated in the abundance/absence of a metabolite: *bb0401* (Figs. 6C and S6C), which encodes a possible glutamic acid transporter[38], *oppAI* (Figs. 6D and S6D), which encodes an oligopeptide binding protein[62,63], *glpF* (Figs. 6E and S6F), which encodes a transmembrane glycerol transporter[38], and *potB* (Figs. 6F and S6G), which encodes a transmembrane protein component of an essential spermidine importer[64,65]. For each gene we detected 5′ RNA fragments that corresponded to 5′ and 3′ ends predicted by RNA-seq, some of which overlapped previously predicted sRNAs[33] (SR0292, SR0242 and SR0541). For the 5′ *bb0401* RNA, a predominant band at the expected size of SR0292 (~202 nt) was observed (Fig. 6C). In contrast, *oppAI* 5′ seemed to harbor multiple short RNAs (Fig. 6D). For both *bb0401* and *oppAI* the levels of the 5′ sRNAs matched the levels of the full-length mRNAs across growth phases. To better resolve the sizes of the short *oppAI* 5′ fragments, we analyzed the same RNA samples on a polyacrylamide gel (Fig. S6E), which documented numerous *oppAI* 5′ fragments, also reflected by the various 3′RNA-seq signals in this region (Fig. 6D). The glycerol uptake facilitator (*glpF*) was previously reported to harbor a long (~195 nt) 5′ UTR that resulted in a premature transcript of the same length[37]. Here, we again detected 5′ derived fragments across growth (Fig. 6E). The levels of the *glpF* 5′ RNAs were predominant in the logarithmic phase sample while the full-length transcript was more abundant later in growth. As for *bgp*, we hypothesize the change in 5′ to full-length *glpF* RNAs could reflect regulatory events.

*potB* is the second gene within a putative polycistronic locus. *B. burgdorferi* is a spermidine auxotroph[66] and the *potABCD* (*bb0642-bb0639*) locus encodes an essential spermidine importer[64]. Specifically, the genes are predicted to encode an ATP-binding protein (PotA; BB0642), two transmembrane proteins (PotB and PotC; BB0641 and BB0640) and a polyamine periplasmic-binding protein (PotD; BB0639)[38,65]. We observed possible premature transcription termination of the *potB* mRNA at a 3′ end that mapped 148 nt into the annotated *potB* ORF (Figs. 6F and S6G). The primary *potB* TSS is internal to the *potA* coding sequencing[37], though total RNA-seq data suggested *potB* could be expressed from a TSS upstream of the *potA* ORF (Fig. 6F). Northern analysis with a probe upstream of the *potB* 3′ end (blue asterisk) documented three predominant products (Fig. 6F). Based on their sizes, we predict these short RNAs originate from the TSS internal to *potA* yielding a ~561 nt transcript (*potB* 5′$_{1-561}$, the distance from the TSS), and two shorter, likely 5′ processed transcripts, at ~228 nt (*potB* 5′$_{334-561}$) and ~77 nt (*potB* 5′$_{485-561}$; SR0541) (Fig. S6G). We confirmed the sizes of the two shorter 5′ fragments by polyacrylamide northern analysis, although other transcripts of lesser abundance were also observed (Fig. S6H). There was a dramatic decrease in the levels of all *potAB*-associated transcripts at the stationary phase timepoints.

In general, the short transcripts that we detected could correspond to *cis*-acting regulators, serve an independent function, and/or exist as the byproducts of regulatory events. These events may be influenced by environmental conditions, for example when the levels of a specific metabolite, such as nucleotides, nitrogen, peptides, carbon sources, or polyamines, are changed. To investigate this hypothesis, we exposed *B. burgdorferi* to spermidine and globally examined transcriptomic changes for 5′ and ORF-internal 3′ ends.

## Spermidine affects levels of ORF-internal 3′ ends

Polyamines play critical roles in cell physiology across all domains of life and their transport and/or utilization are highly regulated. Spermidine in particular has been shown to affect *cis*-acting gene regulation in both eukaryotes and prokaryotes[67]. Three independent *B.*

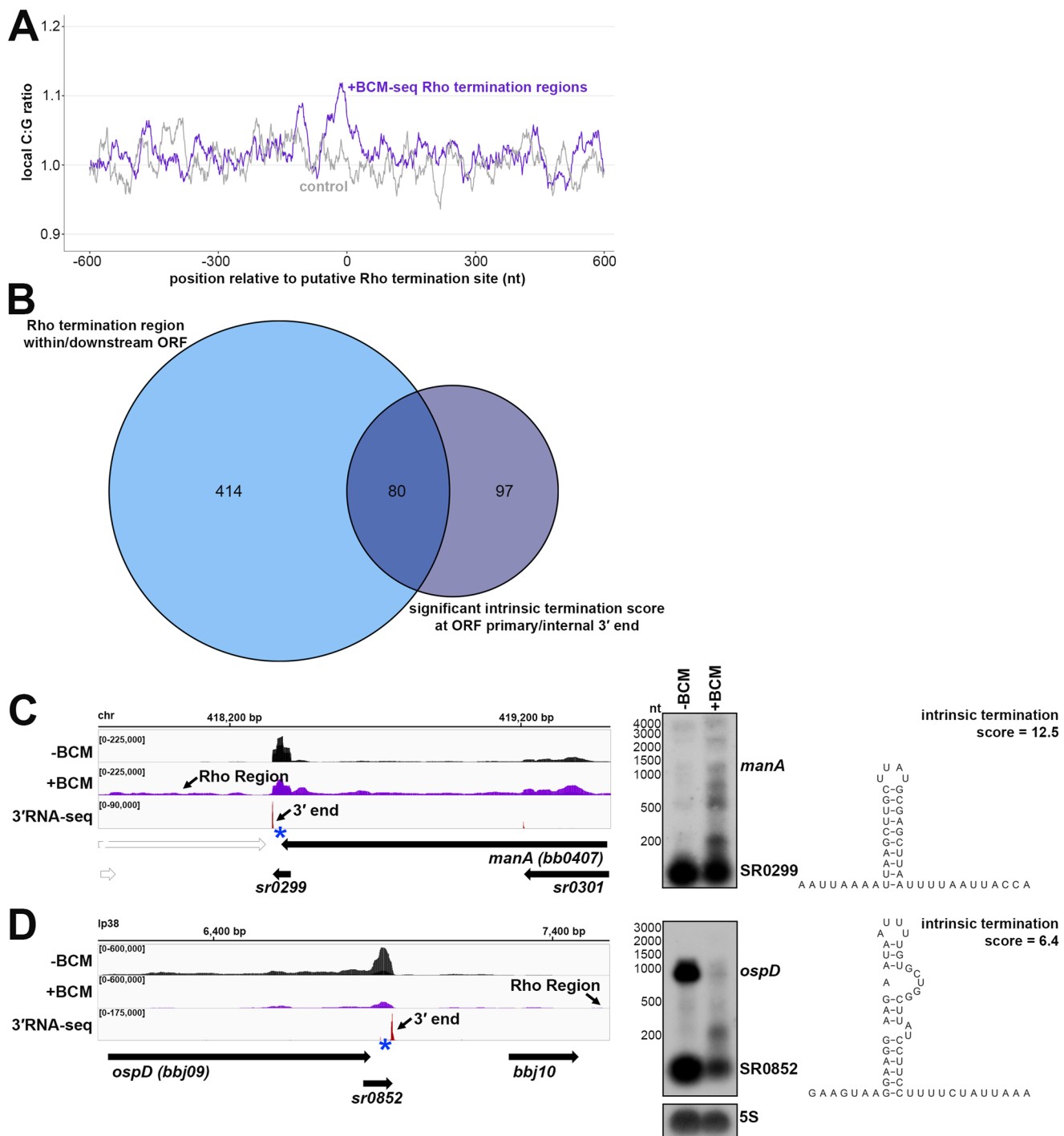

**Fig. 5 | Analysis of putative sites of Rho termination and overlap with intrinsic termination. A** C:G ratio of sequences surrounding predicted Rho termination sites. Nucleotide proportions were calculated by scanning 600 nt upstream and downstream of Rho regions identified in both replicates (Supplementary Data 4) using 25 nt windows. Plotted values represent the average ratios for all 1864 regions (blue), (937 for replicate 1 and 954 for replicate 2). Control plot (gray) represents average C:G ratios calculated in the same manner for 1864 random *B. burgdorferi* B31 genomic positions. **B** Comparison of annotated ORFs with Rho termination regions within or up to 800 bp downstream of their coding sequence to annotated ORFs with a primary or internal 3′ end with a significant intrinsic termination score. RNA-seq browser image (left) and northern analysis (right) for identified Rho regions and putative intrinsic terminators downstream of the (**C**) *manA* mRNA and (**D**) *ospD* mRNA. Browser images display ±BCM-seq and 3′RNA-seq reads as in Fig. 4. Northern analysis was performed on the same blot from Fig. 4 (RNAs were probed sequentially on the same membrane). The 5S blot in panel D was repeated from Fig. 4C/E. Putative terminator structures, modeled from the predicted hairpin structure for the *manA* and *ospD* 3′ end sequence (Supplementary Data 1) are displayed with the corresponding intrinsic termination score.

*burgdorferi* cultures were grown to stationary phase with or without 10 mM spermidine, after which total RNA was isolated and analyzed by RNA-seq. The replicate ±spermidine RNA-seq datasets were highly correlated (Fig. S7A), and the levels of hundreds of transcripts changed in response to spermidine (Fig. S7B; Supplementary Data 6). We focused on the analysis of 5′ and ORF-internal 3′ ends, as we hypothesized that the generation of premature transcripts could be conditionally influenced by excess spermidine.

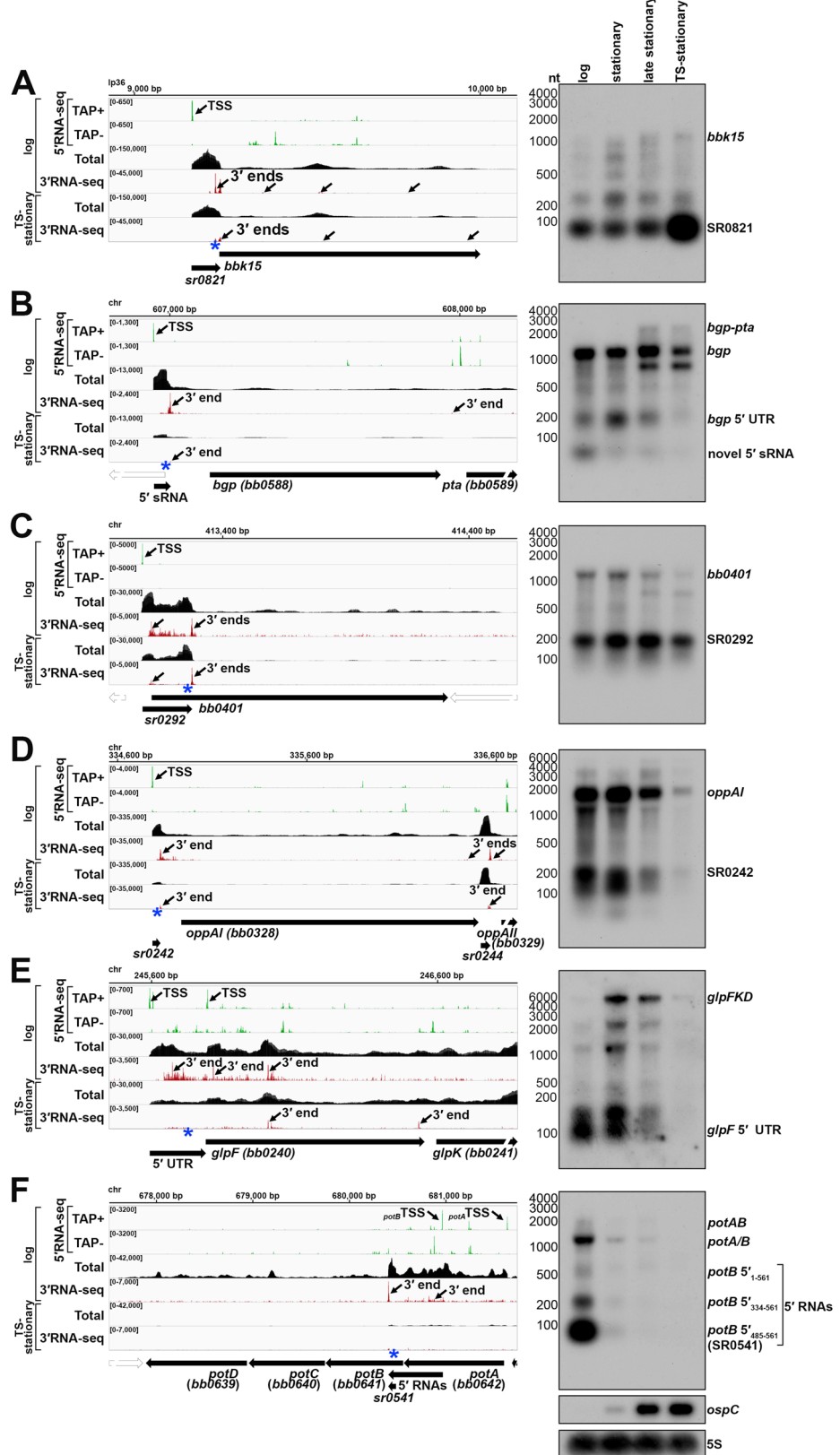

We analyzed the 870 upstream and internal 3′ ends (Supplementary Data 5) and calculated a "spermidine-dependent score", where score of >2.0 is indicative of at least a twofold increase in premature mRNA transcripts in the +spermidine compared to the -spermidine samples. We found 265 genes (30%) to have regions putatively associated with a spermidine-dependent event across at least one

±spermidine replicate (Supplementary Data 5). The majority (75%; 199 3′ ends) were located internal to an annotated ORF; 32 of the 199 had a significant intrinsic termination score and 60 had a significant Rho score.

We examined the *potAB* locus and found strong enrichment of *potB* 5′ RNAs in the +spermidine samples (Fig. 7A). To examine this

**Fig. 6 | Examples of 3′ ends upstream of and internal to ORFs.** RNA-seq browser image (left) and northern analysis (right) for the **A** *bbk15* locus, **B** *bgp/pta* locus, **C** *bb0401* locus, **D** *oppAI/II/III* locus, **E** *glpFKD* locus and **F** *potABCD* locus are shown. Browser images display sequencing reads from logarithmic and TS-stationary phase cells, as in Fig. 2. For northern analysis, cells (PAOO3) were grown at 35 °C, after dilution of the starter culture, to a density of $3.0 \times 10^7$ cells/ml (log) and the culture divided, with part continuing to grow at 35 °C for an additional ~24 h ($3.9 \times 10^8$ cells/ml; stationary) and ~48 h ($4.3 \times 10^8$ cells/ml; late stationary) and the other part temperature-shifted to the bench (~23 °C) for ~48 h and then back to 35 °C for an additional ~48 h, reaching a final density of $3.8 \times 10^8$ cells/ml (TS-stationary).

Cells were lysed at each timepoint, total RNA isolated, separated on an agarose gel, transferred to a membrane and probed for the indicated RNAs. Sample and molecular weight marker (nt) labels are the same for **A–F**; RNAs were probed sequentially on the same membrane. The approximate probe sequence location is indicated by the blue asterisk on the corresponding browser image. The membrane was also probed for *ospC* and 5S as a loading control. Labels for the '*glpF* 5′ UTR' and '*bgp* 5′ UTR' are indicated at the expected UTR size. A previously unannotated sRNA detected within the *bgp* 5′ UTR was denoted 'novel 5′ sRNA' and *potB* 5′ fragments were denoted with coordinates corresponding to their distance from the *potB* TSS (see Fig. S6G).

---

further we isolated total RNA and protein samples from cells grown with or without 10 mM spermidine from the time of sub-culturing, at either logarithmic or stationary phase of growth, best matched to the same cell density (Fig. 7B). Northern analysis revealed an overall decrease in *potAB* transcripts levels in the logarithmic phase +spermidine sample, with no enrichment of the *potB* 5′ RNAs, compared to the logarithmic phase -spermidine sample. Strikingly, the levels of the *potB* $5′_{1-561}$ and SR0541 RNAs increased dramatically in the stationary-phase cells treated with spermidine, with a concomitant decrease in full-length *potA/B* mRNAs. To delineate which bands corresponded to full-length *potA*, *potB*, and *potAB* transcripts, we carried out northern analysis with multiple probes and concluded that the genes can be transcribed together (*potAB*) or as separate RNAs of approximately the same size (*potA/B*) (Supplementary Fig. S6I). We also analyzed protein lysates from the same experiment (analysis of Fig. 7B samples) using antibodies to native PotA[65]. This immunoblot showed PotA protein levels also decreased with the addition of 10 mM spermidine in both logarithmic and stationary samples (Fig. S8A).

We observed a defect in spirochete growth and ~16 h delay for cells cultured with 10 mM spermidine when collecting samples for northern analysis (Fig. 7B). Therefore, we also examined the *potB* 5′ RNA levels when *B. burgdorferi* cells were grown to logarithmic phase and then exposed to 0, 5, or 10 mM spermidine for 48 h. Under these conditions, all samples reached the stationary phase at approximately the same time. RNA was isolated and northern analysis was performed for the *potB* 5′ RNAs (Fig. 7C). There was no increase in the *potB* 5′ RNAs with 5 mM spermidine, though *potA/B* full-length mRNAs decreased. As before, the levels of the *potB* $5′_{1-561}$ and SR0541 RNAs increased with 10 mM spermidine, while the levels of the full-length transcripts decreased. Based on these observations, we hypothesized that excess spermidine enhances the generation of premature *potB* RNAs and causes downregulation of the transport system under conditions of excess spermidine.

We selected two other examples of genes harboring spermidine-dependent, ORF-internal 3′ ends for northern analysis. *bbo213* encodes a hypothetical protein and is co-transcribed with *efp* encoding elongation factor protein P, which, in other organisms, has been shown to aid translation of proteins with polyproline motifs[68,69]. *cdd* (*bb0618*) encodes a cytidine deaminase[38]. We observed a 3′ end that mapped 259 nt into the annotated *bb0213* ORF (Fig. S7A) and a 3′ end that mapped 289 nt into the annotated *cdd* ORF (Fig. S7D). Northern analysis detected an enrichment of premature-*bb0213* and -*cdd* transcripts when spirochetes were grown with 10 mM spermidine (Fig. S7E and S7F). Our data suggest that spermidine-mediated regulation is broadly affecting instances of internal 3′ ends and the accumulation of truncated mRNAs.

### Spermidine-dependent regulation of potA/B and an ORF-internal RNA

To investigate if the downregulation of *potA/B* occurs post-transcription initiation we created transcriptional and translational luciferase reporter fusions using 200 bp upstream of the primary *potB* TSS (P*potB*, transcriptional fusion) or the same sequence with *potB* $5′_{1-561}$ + 33 nt downstream of the premature 3′ end (P*potB*+*potB* $5′_{1-594}$,

translational fusion), which fuses the first 61 amino acids of PotB in-frame with luciferase (Fig. S6G). We assayed both constructs, along with vector control, for stationary phase cells treated with spermidine (as in Fig. 7C). Luciferase units were normalized to the number of cells in each sample (raw data provided in Source Data) and are reported as a percentage relative to the 0 mM spermidine samples (Fig. 7D). *potB* promoter activity decreased with the addition of either 5 or 10 mM spermidine. There was no significant difference in the luciferase activity between the two concentrations of spermidine. The luciferase activity from the *potB* translational fusion decreased similarly for cells exposed to 5 mM spermidine. However, a stronger, significant decrease (~70% reduction) in luciferase activity was observed with the translational fusion exposed to 10 mM spermidine. These reporter-fusion data corroborated results from our northern analysis and suggest that *potB* expression is regulated at both the promoter and after transcription initiation.

We were curious if the spermidine-*potB* $5′_{1-561}$ effect could still occur with another promoter. Thus, we used the same translational luciferase reporter fusion but replaced the *potB* promoter with the *potA* promoter by incorporating 200 bp upstream of the *potA* TSS (P*potA*+*potB* $5′_{1-594}$, translational fusion). In the 5′RNA-seq data, the number of TSS reads for P*potA* and P*potB* are similar (Figs. 6F and 7A). We assayed the hybrid construct (P*potA*+*potB* $5′_{1-594}$), along with a vector control and a *potA* transcriptional reporter fusion control (P*potA*, transcriptional fusion) under the same experimental conditions as before. As for the *potB* promoter, *potA* promoter activity decreased with the addition of spermidine, with only a marginal difference between 5 and 10 mM spermidine (Fig. 7D). However, we again observed a stronger decrease in luciferase activity (~75% reduction) when the hybrid translational fusion was exposed to 10 mM spermidine compared to 5 mM spermidine. This experiment demonstrates a *potB* 5′ spermidine-dependent effect, regardless of the promoter which drives its expression. We noted that the *potB* $5′_{485-561}$ (SR0541) RNA sequence harbors a predicted hairpin secondary structure and a series of rare codons, one being a proline, immediately upstream the ORF-internal 3′ end (Fig. S8B and S8C). To test if this sequence could be important for the effects of spermidine, we codon optimized the *potB* ORF within the translational fusion (Fig. S8B), which also disrupted the RNA secondary structure. We no longer observed spermidine-dependent effects on luciferase activity for this construct (Fig. S8D). Collectively, these data suggest that in the stationary phase excess spermidine represses *potAB* transcription. Additionally, RNA levels are downregulated post-transcription initiation leading to *potB* 5′ RNA fragments and further reduction of the transport system.

### potB 5′ RNAs expressed across the tick-mouse enzootic cycle

Spermidine levels vary across mammalian tissues, and spermidine is the only polyamine available in the midgut of nymphal ticks[64,70]. We hypothesized that levels of RNAs associated with premature termination and/or other regulatory events could change based on the spirochete's environment, reflective of a physiological role. To obtain more information about the truncated *potB* transcript, we carried out reverse transcription (RT)-qPCR using primers within SR0541 to measure the levels of *potB* 5′ RNAs across the tick-mouse enzootic cycle

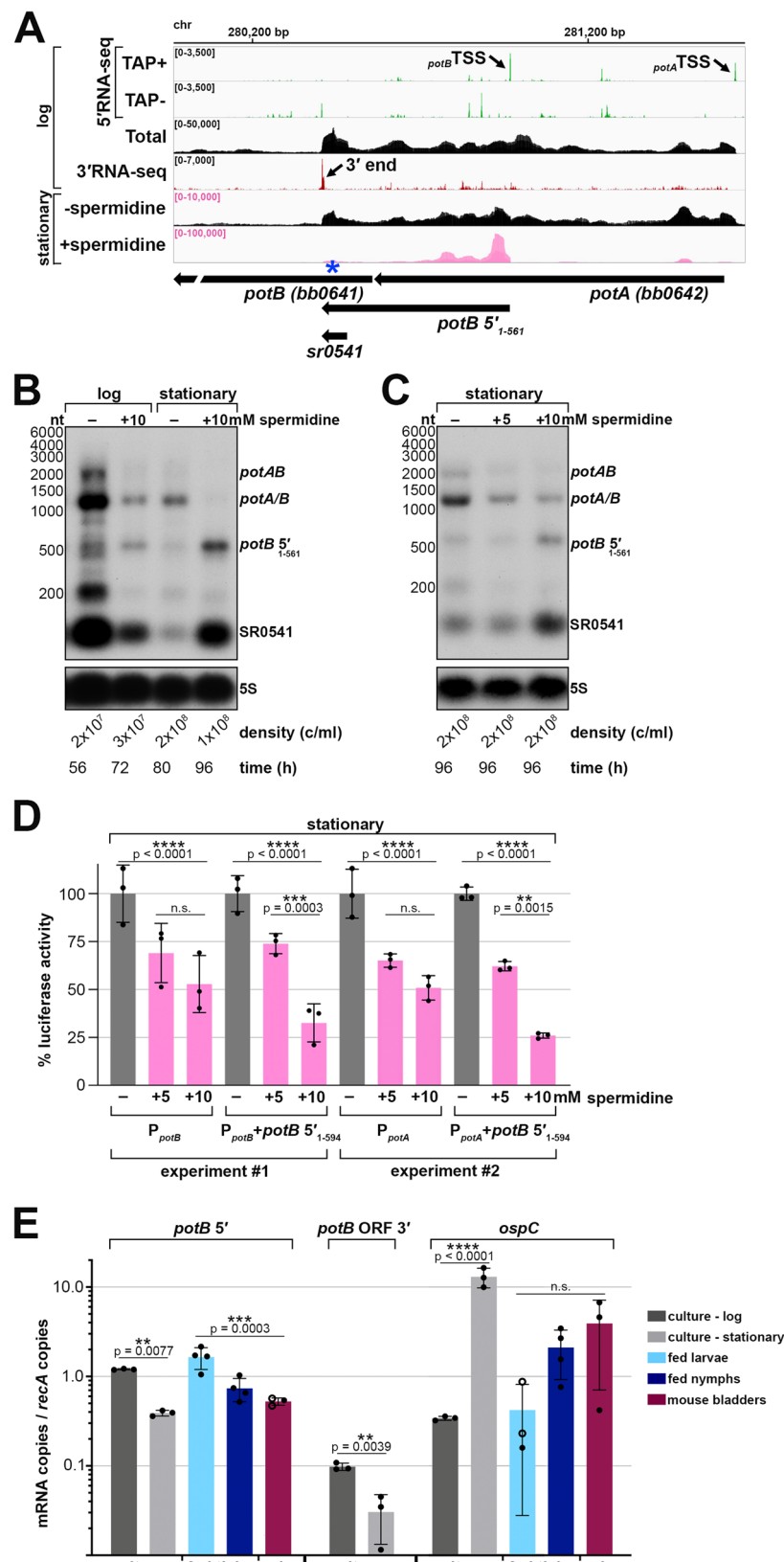

(Fig. 7E). As a control, we probed the same cDNA samples for *ospC* levels, which, as expected, were higher in culture-derived stationary phase samples than logarithmic phase and increased in mice compared to fed ticks. Consistent with the *potB* northern analysis (Fig. 6F), *potB* 5′ levels were lower in the stationary phase cDNA samples compared to the logarithmic samples (Fig. 7E). *potB* 5′ levels were ~7-fold

higher in fed larvae compared to infected mouse tissues (Fig. 7E). In contrast, RT-qPCR carried out with primers targeting a region downstream of the *potB* premature transcript was below the limit of detection in the environmental samples. This expression pattern indicates that *potB* 5′ levels change across the tick-mouse enzootic cycle, which could reflect a regulatory response to different

**Fig. 7 | Spermidine affects levels of potAB and an ORF-internal RNA. A** RNA-seq browser image of the *potAB* locus. Browser image display total-, 5′ and 3′ RNA-sequencing reads from logarithmic phase as in Fig. 2, with the addition of stationary phase ±spermidine total RNA-seq. Stationary ±spermidine tracks represent an overlay of three biological replicates; spermidine treatment reads are colored pink. Read count ranges are shown in the top left of each frame. Scaling (pink text) for +spermidine is different from -spermidine to better visualize the -spermidine reads. **B** Northern analysis of effects of spermidine on *potAB* transcripts across growth. Cells (PA003) were diluted into BSK-II or BSK-II + 10 mM spermidine, adjusted to pH 7.8. Samples were harvested at the indicated time points to best match cell density (indicated below the 5S blot). **C** Northern analysis of effects on *potAB* transcripts when exposed to 0, 5, and 10 mM spermidine. One starter-culture of *B. burgdorferi* (PA003) was diluted into 3 individual BSK-II cultures and grown at 35 °C to a density of ~ 3.0 × 10^7 cells/ml. Cell pellets were collected by centrifugation and resuspended in BSK-II, BSK-II + 5 mM spermidine, or BSK-II + 10 mM spermidine, adjusted to pH 7.8. Cultures incubated at 35 °C for 42 h all reached a similar final cell density (indicated below the 5S blot), when samples were collected and cells lysed. For both panels B and C, total RNA was isolated, separated on an agarose gel, transferred to a membrane and probed for *potAB* transcripts and 5S as a loading control. The approximate probe sequence location is indicated by the blue asterisk on the browser image in panel **A**; size markers are indicated. Panels **B** and **C** are representative experiments from three independent experimental replicates, all showing the same results. **D** Luciferase analysis of fusions harboring the *potB* 5′ sequence. *B.*

*burgdorferi* containing transcriptional ($P_{potB}$; strain PA345, $P_{potA}$; strain PA343) or translational ($P_{potB}+potB$ 5′$_{1-594}$; strain PA223, $P_{potA}+potB$ 5′$_{1-594}$; strain PA381) fusions to luciferase were grown and exposed to spermidine as in panel **C**. Luminescence measurements were normalized to the cell density in each sample by $OD_{600}$. Each data point (black circles) represents an independent biological replicate ($n=3$) as the percent luciferase activity relative to the average normalized luciferase units in the -spermidine condition for each construct. Bars represent the average luciferase activity across all biological replicates ± standard deviation. Results from two independent experiments are shown. Average activity across samples from both experiments were compared by one-way ANOVA with Šídák's multiple comparisons test, GraphPad Prism 9.4.1 (n.s., not significant), see Source Data file for all measurements and full statistical analysis. **E** Reverse transcription (RT)-qPCR analysis of *potB* 5′, *potB* ORF 3′ (downstream of the *potB* internal 3′ end) and *ospC* mRNA levels in culture compared to infected tick and mouse bladder tissue samples. Copy numbers for each RNA were normalized to *recA* mRNA copies. Circles correspond to independent biological replicates ($n=3$); filled circles represent the average of two or three technical replicates while open circles indicate that only one of the technical replicates for that sample amplified (see Source Data file). Bars represent the average across all biological replicates ± standard deviation. Expression levels across sample groups were compared by one-way ANOVA with Šídák's multiple comparisons test or t test (*potB* ORF 3′ samples), GraphPad Prism 9.4.1 (n.s., not significant), see Source Data file for full statistical analysis.

---

spermidine concentrations in the various environments. Generally, instances of premature termination and RNA-mediated regulation are likely important during *B. burgdorferi* infection of ticks and mammals.

## Discussion

We took an RNA-seq approach to globally map RNA 3′ boundaries in *B. burgdorferi*. Given that the strain and logarithmic growth conditions used for 3′ RNA-seq and total RNA-seq in this study match those of the previous 5′ RNA-seq analysis[37], the combined sets represent a valuable resource for examining the *B. burgdorferi* transcriptome. Using these data together, one can simultaneously distinguish TSSs and 5′ processed ends, examine total sequencing coverage of RNAs, detect 3′ ends, and identify instances of possible regulation (see Methods for links to interactive browsers).

We also performed 3′ RNA-seq and total RNA-seq on temperature-shifted (TS) stationary phase samples. While no growth condition in BSKII culture medium fully mimics environmental conditions, this approach has been utilized to robustly induce *rpoS*[39,40,71]. We observed differences in the levels of multiple transcripts when comparing late stationary samples to TS-stationary samples (Fig. 6). This suggests a temperature shift has different consequences on gene expression than growth to the stationary phase alone.

Our datasets are critical for improving gene annotations of *B. burgdorferi* and revealed complex and overlapping transcription indicative of nuanced regulation. We updated previous gene annotations using the 5′- and 3′ RNA-seq data. We note that this list (Supplementary Data 3) is generated largely by algorithmic detection. In many cases we believe this annotation accurately reflects biological products, even in complex gene arrangements. For example, our algorithms identified overlapping UTRs and internal/5′/3′-derived sRNAs within mRNAs, which largely reflects the annotations we would predict based on visual inspection of the RNA-seq data. However, in some instances, an abundance of sequenced TSSs and/or 3′ ends create challenges in distinguishing transcriptional units. As with any high-throughput method for identifying RNA-seq peaks, genes of interest should be individually examined for features that could be missed by an algorithmic curation. For example, our algorithm only identified the most dominant and primary 3′ end in the *ospAB* operon, but visual inspection of 3′ RNA-seq and northern analysis documented the presence of other internal 3′ ends (including those associated with SR0884 and a novel *ospA* 5′ derived sRNA). In a few cases, 3′ ends were absent from the RNA-seq datasets but observed by northern analysis. A lack of 5′ or

3′ end identification could be a result of sequencing depth or biases, such as reduced efficiency during RNA adaptor ligation or cDNA amplification. It is also possible a subset of RNA ends is missed due to RNA degradation during library preparation. Nevertheless, we think that our data and list of 3′ ends (Supplementary Data 1) are reflective of most RNA ends present under the conditions where the RNA was isolated.

We resolved some controversies about mRNA annotation and uncovered several examples of operon misannotation. For example, there has been controversy about the genetic organization of *bgp* and the downstream *pta* genes. One study suggested the genes were expressed independently[72], while two other studies predict the genes are co-transcribed[59,73]. Our northern analysis (Fig. 6B) suggested the transcriptional units are growth-phase dependent. Across all growth phases, *bgp* was primarily transcribed alone, but a band corresponding to the size of the *bgp-pta* RNA was detected in samples isolated from late stationary phase spirochetes. We want to point out that transcriptome complexities can be overlooked by RT-qPCR analysis. Several groups used RT-qPCR to test for co-transcription of genes by amplifying across annotated gene gaps: *bb0360-bb0364*[74]; *bb0215-bb0218*[75]; *bb0404-bb0406*[76]. However, multiple TSSs were previously detected in these examples of predicted gene operons[37], and we now observe multiple 3′ ends. Finally, some transcripts, especially ORF-internal RNAs, could be missed by total RNA-seq analysis alone, likely contributing to the substantial differences between our work and a prior total RNA-seq study that predicted gene terminators[16].

We also updated sRNA annotations. For some sRNAs it is more challenging to define sRNA boundaries, as they can exist and function in multiple forms. We visually examined each predicted sRNA, using the pattern and abundance of sequencing reads across all our RNA-seq datasets for annotation, in some cases listing multiple RNA ends for one sRNA. Differences with previous datasets could be attributed to the use of size-selection RNA-seq[33], which creates a bias for sequencing small transcripts and could enrich for low-abundance processed/degraded products, or the use of total RNA-seq alone[16], where it may be difficult to distinguish sRNA signals among the reads from longer transcripts. In some cases, multiple sRNAs had been annotated in one location, but our 5′ and 3′ end annotation suggested only one sRNA, so we combined the annotations. In other cases sRNAs did not meet the criteria for algorithmic curation. For example, previous sRNA annotations for SR0694 were in proximity but not within our 75 nt window of a detected 5′ and 3′ end. So, while we associated the *panF* ORF-internal

sRNA with SR0694 in Fig. 4D, this sRNA was absent from Supplementary Data 2. In our annotation files, we also removed all tRNAs previously assigned sRNA nomenclature[33]. We detected several novel transcripts, particularly small transcripts, some derived from genes required for *B. burgdorferi* infectivity. Some sRNAs are primarily detected in a particular growth phase, like SR0754 (Fig. 2C) which lacked robust 3′ end detection in the logarithmic phase samples. In the future, we aim to systematically search for additional sRNAs in our RNA-seq datasets and continue to improve sRNA annotation.

Our data also suggest underappreciated regulation at the level of the transcriptome. The majority of our identified 3′ ends were located within annotated genes (59.1%), reminiscent of the previous observation that 63.0% of TSSs mapped internal to annotated *B. burgdorferi* ORFs[37]. One ORF-internal 3′ end had been previously reported for *bmpB* (*bb0382*)[17], which we also detected in concurrence with an ORF-internal sRNA (SR0274). Recently, a high percentage of ORF-internal 3′ ends also was reported for *Mycobacterium tuberculosis*[77]. The presence of ORF-internal RNAs may reflect regulated transcription termination or RNA processing which could alter transcript sizes based on growth stage or the environment of the bacterium, changing expression of the encoded gene products.

Moving forward, our list of gene annotations (Supplementary Data 3) should be continually updated. In general, we established that multiple types of RNA-seq data together with northern analysis allows the most confident delineation of transcriptional units. Additional ribosome profiling studies to map the boundaries of translation[78–80] are needed to further refine protein-coding annotations.

Transcription termination has been well-studied in model bacteria like *E. coli* and *B. subtilis*, but not comprehensively analyzed in *B. burgdorferi*. Using our datasets, we classified 3′ ends as possible intrinsic terminators if their upstream sequence had features of canonical intrinsic terminators. When we compared our analysis of *B. burgdorferi* 3′ ends with similar data for *E. coli*, *P. aeruginosa* and *B. subtilis*, we observed the greatest similarity between the predicted intrinsic terminators in *E. coli* (30.0% of the 884 detected 3′ ends) and *B. burgdorferi* (18.9% of the 1,333 detected 3′ ends). Both organisms had a similar putative intrinsic structures and termination logos. It is somewhat surprising that there is not a stronger enrichment of G/C content surrounding *B. burgdorferi* 3′ ends, but given the A/T rich genome, perhaps limited G/C content in terminators is sufficient for the formation of stable secondary structures. In contrast, *P. aeruginosa*, which had the highest genomic G/C content of the organisms tested, had a strong enrichment of G/C content at 3′ ends. Of note, our classification of intrinsic termination is based on a previously described quantitative model of known *E. coli* and synthetic intrinsic terminators[48]. There could be novel features of *B. burgdorferi* terminators that prevent their "intrinsic" classification. Most instances of *B. subtilis* intrinsic termination require the protein factors NusA and/or NusG to facilitate efficient transcription termination[13,14]. It is worth testing if the *B. subtilis* model of protein factor-mediated intrinsic termination is applicable to bacteria where significantly fewer canonical intrinsic terminators are predicted. Genes encoding NusA and NusG are both annotated in *B. burgdorferi*[38], but have not been studied, although NusA was found to co-purify with *B. burgdorferi* RNA polymerase[81]. Possibly, yet-uncharacterized termination factors are also involved in *B. burgdorferi* intrinsic termination. Further work, including in vitro assays using purified *B. burgdorferi* RNA polymerase, is required to define the "rules" of intrinsic termination in this pathogen.

We documented that Rho is involved in *B. burgdorferi* termination using bicyclomycin treatment and mapped "Rho regions" by RNA-seq, the first report of Rho-dependent transcripts in *B. burgdorferi*. We observed 623 likely instances of Rho-mediated termination downstream of and within annotated ORFs, higher than the number of predicted intrinsic terminators. There is some controversy about when

Rho associates with RNA and the RNA polymerase elongating complex, and the function/significance of C-rich Rut sites[82–84]. Our analysis suggested a limited increase in local C: G content preceding Rho termination regions in *B. burgdorferi*. We observed that some Rho regions were sharply defined by a dominant 3′ end, while others were stuttered with multiple 3′ ends. A well-defined 3′ end does not necessarily mean it arose from intrinsic termination, as even Rho-dependent 3′ ends may be preceded by regions of structured RNA or a termination event could be robustly 3′ processed to a specific nucleotide. We observed that 45.2% of predicted mRNA intrinsic terminators have downstream Rho regions. An overlap between intrinsic and Rho-mediated termination was also observed in *E. coli*[12] and *B. subtilis*[13]. We carried out northern analysis on two *B. burgdorferi* examples, *manA*, and *ospD*, both with significant intrinsic terminator scores and a downstream Rho region. In both cases, we observed an overall decrease in the full-length mRNAs and an increase in shorter mRNA-fragments, when Rho was inhibited. This would suggest that both Rho and 3′ RNA secondary structures are required for appropriate termination and stability of the full-length transcript. Shorter RNA fragments may be generated as a byproduct of RNA turnover when termination occurs. It is also possible that this could serve as a regulatory mechanism, where the abundance or ability of Rho to bind could be titrated to control the length of an RNA produced under a specific condition.

Many other detected 3′ ends in our dataset were not classified as either intrinsic or Rho-dependent terminators and could be generated by novel mechanisms of termination or result from RNA processing. Limited studies have examined RNA processing and turnover in *B. burgdorferi*[85,86]. The spirochete is predicted to encode endoribonucleases orthologs of RNase Y (*bb0504*), RNase III (*bb0750*), RNase HII (*bb0046*), RNase P (*bb0441*), RNase Z (tRNA 3′ endonuclease; *bb0755*), RNase M5 (5S maturation nuclease; *bb0626*), YbeY (*bb0060*) and exoribonuclease orthologs of Oligoribonuclease (*bb0619*) and PNPase (*bb0805*)[38,85,87]. Processed RNA ends often result in/depend on RNA secondary structures to stabilize the transcript. For example, the 3′ to 5′ exoribonuclease PNPase catalyzes nonspecific removal of 3′ ribonucleotides but is blocked by dsRNA structures formed at 3′ ends[51]. In some cases, dsRNA structures with low energies of dissociation may not uniformly protect the transcript, resulting in the clusters of minor 3′ ends that we observe. Transcription termination and the stability of mRNA 3′ ends are inherently variable processes impacted by multiple factors that can change across growth. Understanding the basic biological process of transcription termination and the contributing factors is important for understanding these regulatory processes.

We focused on the abundant instances of 3′ ends in 5′ UTRs and ORFs, as we propose that their identification is an effective approach to discovering novel regulatory elements. Classically, mRNA internal regulators, such as riboswitches and attenuators, have been identified by serendipity, studies of individual genes, or searches for conserved RNA structures[19], but these approaches may miss regulatory RNA elements, especially in organisms like *B. burgdorferi* where the functions of many genes are unknown and there is limited conservation compared to well-studied bacteria. Given that 3′RNA-seq is a sensitive, relatively unbiased, and genome-wide approach, we think it can provide evidence for RNA-mediated regulation for further study. Several long UTR sequences, such as *pncA*, *bgp*, and *glpF*, were found to harbor 3′ ends and stable small RNA fragments. In other cases, 3′ ends and sRNAs were detected within the start of a coding sequence as for *bb0401* and *potB*. We hypothesize that a subset of these instances may reflect the presence of *cis*-acting regulation such as RNA thermometers and riboswitches. Bioinformatic predictions of riboswitches have suggested their presence in Spirochaetes[88], yet the sole riboswitch characterized to date is a thiamine pyrophosphate (TPP) riboswitch in *Treponema denticola*[89]. We expect there are unique regulatory mechanisms for the numerous 3′ ends we detected.

To explore the utility of our approach to finding novel regulation, we investigated the effect of spermidine on transcripts detected. The polyamine spermidine globally affects translational fidelity by post-translationally modifying the eIF5A translation factor in eukaryotes, particularly for the translation of poly-proline residues[90]. The global impact of spermidine in bacteria is less well studied. There are several examples in both bacteria and eukaryotes where translation of an upstream open reading frame (uORF) controls the expression of a downstream polyamine biosynthetic or regulatory gene. In *E. coli*, spermidine induces ribosome stalling at a uORF in the 5′ leader of the *speFL* mRNA[91,92] and affects the translation of a uORF upstream the *E. coli mdtJI* mRNA[12], encoding a spermidine exporter. In eukaryotic examples, when polyamine levels are high, there is enhanced ribosome occupancy on the uORF resulting in lower expression of the downstream gene[93–95].

We discovered a prematurely generated fragment of *B. burgdorferi potB* that increased in response to spermidine alongside down-regulation of the full-length mRNA. The exact mechanism for this effect deserves additional study. However, given the overlap with the *potA* and *potB* coding sequence and the examples in other organisms, we hypothesize that *potB* 5′ RNAs could reflect the modulation of translation efficiency coupled with premature termination or mRNA cleavage. Our transcriptome-wide mapping of *B. burgdorferi* 3′ ends in response to spermidine also documented other examples of increased 5′ fragments at the start of coding sequences.

Our work has led to new insights into the fundamental process of termination of transcription in *B. burgdorferi*, spirochete gene regulation, and thus the biology of Lyme disease. We documented an abundance of potential *cis*- and *trans*-acting regulatory elements that set the stage for future projects to characterize RNA-mediated gene regulation in *B. burgdorferi*.

## Methods
### Bacterial strains and plasmids
Derivatives of infectious wild-type *B. burgdorferi* strain B31 A3 lacking cp9[96], Adams lab strain number PA003, were used for all experimental studies. All strains, plasmids, and oligonucleotides used are listed in Supplementary Data 7. Engineered plasmid inserts were verified by sequencing.

The luciferase reporter plasmids were generated in NEB 5α *E. coli*, grown in LB broth or on LB agar plates containing 300 µg/ml spectinomycin, prior to transformation into *B. burgdorferi* A3-68Δ*bbeO2*, which lacks cp9, lp56 and gene *bbeO2* on lp25[97]. The transcriptional luciferase reporter fusion plasmids were constructed by PCR-amplifying 200 bp upstream the *potB* or *potA* TSS[37] using *B. burgdorferi* B31 (PA003) genomic DNA and the primers PA553 + PA554 or PA535 + PA536, respectively. The promoter sequences were ligated into PCR-amplified (PA480 + PA481) pJSB161 vector[98] using the NEBuilder HiFi DNA assembly master mix, according to the manufacturer's instructions (NEB). The *potB* translational luciferase reporter fusion plasmid was constructed by PCR-amplifying (primers PA400 + PA402) 200 bp upstream the *potB* TSS along with *potB* 5′_{1-561} + 33 nt downstream of the dominant *potB* ORF-internal 3′ end (P_{potB}+*potB* 5′_{1-594}), which fuses the first 61 PotB amino acids in-frame with luciferase, and cloned into the pJSB161 vector cut with BlgII and BanII restriction enzymes. The codon-optimized *potB* translational luciferase reporter sequence was synthesized (GenScript) and then PCR-amplified (PA402 + PA531), prior to ligation into PCR-amplified (PA563 + PA590) pJSB161 vector using the NEBuilder HiFi DNA assembly master mix. The P_{potA}+*potB* 5′_{1-594} translational luciferase reporter fusion plasmid was constructed by ligating PCR-amplified products for P_{potA} (primers PA601 + PA602) and *potB* 5′_{1-594} (primers PA603 + PA594), which harbored an overlapping sequence into PCR-amplified (PA563 + PA590) pJSB161 vector using the NEBuilder HiFi DNA assembly master mix.

### Growth conditions
*B. burgdorferi* were cultivated in liquid Barbour-Stoenner-Kelly (BSK) II medium supplemented with gelatin and 6% rabbit serum[32] and plated in solid BSK medium as previously described[99]. *B. burgdorferi* cultures were grown at 35 °C with 2.5% CO_2. Spirochete starter cultures were grown from a freezer stock, cell density enumerated by darkfield microscopy, and subcultured once by diluting spirochetes to $1 \times 10^5$ cells/ml for all experiments. Strains harboring luciferase reporter plasmids were grown in the presence of 50 µg/ml streptomycin. Temperature-shifted stationary cultures were grown to mid-logarithmic phase ($3.0 \times 10^7 – 5.0 \times 10^7$ cells/ml) at 35 °C, temperature shifted to the bench (~23 °C) for 48 h, and then temperature shifted back to 35 °C for 48 h resulting in a stationary culture ($2.0 \times 10^8 – 3.0 \times 10^8$ cells/ml). Growth curves were performed by diluting spirochetes to $1 \times 10^5$ cells/ml and monitoring cell growth at the indicated time points by darkfield microscopy enumeration.

### RNA isolation
45 ml of a *B. burgdorferi* culture was collected by centrifugation, washed once with 1X PBS (1.54 M NaCl, 10.6 mM KH_2PO_4, 56.0 mM Na_2HPO_4, pH 7.4), and pellets snap frozen in liquid N_2. RNA was isolated using TRIzol (Thermo Fisher Scientific) exactly as described previously[100]. RNA was resuspended in DEPC H_2O and quantified using a NanoDrop (Thermo Fisher Scientific).

### 3′RNA-seq
Three biological replicates of *B. burgdorferi* B31 (PA003) were grown in BSKII to mid-logarithmic phase ($3.75 \times 10^7 – 5.5 \times 10^7$ cells/ml) at 35 °C, the culture split for RNA isolation or temperature shifted (TS) to bench (~23 °C), and then back to 35 °C and grown to stationary phase (~$3.2 \times 10^8$ cells/ml). RNA was extracted as described above, resulting in logarithmic and TS-stationary samples. RNA was analyzed using an Agilent 4200 TapeStation System to check the quality. RNA was prepared for sequencing as described previously[12]. Any contaminating DNA in the samples was removed by treating 15 µg of RNA with 10 U of DNase I (Roche) for 15 min at 37 °C in the presence of 80 U of recombinant RNase inhibitor (Takara Bio). Next, RNA was purified by mixing the sample with an equal volume of phenol stabilized:chloroform:isoamyl alcohol (25:24:1) and centrifugation at maximum speed in Heavy Phase Lock Gel tubes (5 PRIME). A volume of chloroform equal to the original sample volume was added to the same Heavy Phase Lock Gel tubes and spun again. The aqueous layer was removed and ethanol precipitated. RNA pellets were reconstituted in 10 µl DEPC H_2O and analyzed using an Agilent 4200 TapeStation System to ensure DNase-treated RNA was of high quality. 3′RNA-seq libraries were prepared using a modified version of the RNAtag-seq methodology[41] based on the previously published Term-seq methodology[18]. 1.5 µg of DNA-free RNA was first ligated at the 3′ ends with 150 µM barcoded oligonucleotide adapters, which were 5′ phosphorylated and dideoxycytidine 3′ terminated (Supplementary Data 7). RNA and 3′ adapters were incubated at 22 °C for 2.5 h with 51 U of T4 RNA Ligase I (NEB) and 12 U of recombinant RNase inhibitor (Takara Bio) in 1X T4 RNA Ligase Buffer (NEB), 9% DMSO, 20% PEG 8000, and 1 mM ATP. 3′ ligated RNA was cleaned by incubating with 2.5X volume of RNAClean XP beads (Beckman Coulter) and 1.5X volume of isopropanol for 15 min before separation on a magnetic rack. Bead-bound RNA was washed with 80% ethanol, air-dried, and resuspended in DEPC H_2O. RNA-containing-supernatants were removed and the same RNAClean XP bead cleanup protocol was repeated, with a final DEPC H_2O elution of 9.5 µl. RNA was fragmented by incubating 9 µl of cleaned-up RNA with 1X Fragmentation Reagent (Invitrogen) for 2 min at 72 °C, followed by addition of 1X Stop Solution (Invitrogen). Samples were stored on ice following individual fragmentation of each sample. Fragmented-RNA was pooled together and cleaned using the RNA Clean and Concentrator-5 kit (Zymo) according to the manufacturer's instructions. Library

construction continued following the bacterial-sRNA adapted, RNAtag-seq methodology starting at the rRNA removal step[101]. 3′RNA-seq libraries were analyzed on a Qubit 3 Fluorometer (Thermo Fisher Scientific) and an Agilent 4200 TapeStation System prior to paired-end sequencing using the HiSeq 2500 system (Illumina).

## Identification of 3′ ends from 3′RNA-seq

Raw sequence reads were processed using lcdb-wf (lcdb.github.io/lcdb-wf/) according to the following steps as previously performed[12]. Raw sequence reads were trimmed with cutadapt 2.3[102] to remove any adapters while performing light quality trimming with parameters "-a AGATCGGAAGAGCACACGTCTGAACTCCAGTCA -q 20 –minimum-length=25." Sequencing library quality was assessed with fastqc v0.11.8 with default parameters and multiqc v1.10. The presence of common sequencing contaminants was evaluated with fastq_screen v0.13.0 with parameters "–subset 100000 –aligner bowtie2." Trimmed reads were mapped to the *B. burgdorferi* reference genome (GCF_000008685.2_ASM868v2) using BWA-MEM. Multimapping reads were filtered using samtools[103]. Uniquely aligned reads were then mapped using subread featureCounts v2.0.1 with default parameters to gene features made from the union of NCBI *B. burgdorferi* B31 gene annotations GCF_000008685.2 and the updated sRNA annotation described below. BedGraph files were generated using deepTools[104] on reads from each strand separately.

An initial set of termination peaks was called per sample on the uniquely aligned reads using a novel signal processing approach combined with a statistically-informed method of combining multiple replicates. Briefly, the scipy.signal Python package was used to call peaks on each replicate in a manner that handled high, sharp peaks as found in 3′RNA-seq data, using the scipy.signal.find_peaks function with a width of (1, None), a prominence of (None, None), and a relative height of 0.75. Peaks for each replicate were then combined using the IDR framework[105] into a set of peaks that were reproducible across replicates. This code can be found at https://github.com/NICHD-BSPC/termseq-peaks and can be used in general for 3′RNA-seq (Term-seq) peak-calling in other bacteria. Termination peaks were subsequently curated according to the following criteria. The single-bp peak coordinate was set to the strongest signal nucleotide within the boundary of the initial broader peak using multiBigWigSummary from deep-Tools 3.2.1. The most downstream position, relative to the peak orientation, was chosen when several positions were equally strong. Scores from peaks within a distance of up to 100 bp were assessed to select the peak with the highest score among the cluster for further analysis. These curated peaks were used for all analysis herein (Supplementary Data 1).

## Classification of 3′ ends

Each peak was assigned to one or more classes, based on the intersect function of Bedtools 2.27.1[106], ran via pybedtools v0.8.0[107] and divides the peaks into: Primary (3′ peaks located on the same strand either within 100 bp downstream of the 3′ end of an annotated mRNA ORF, tRNA, rRNA or sRNA with the highest score), Antisense (3′ peaks located on the opposite strand of an annotated mRNA ORF, tRNA, rRNA or sRNA within 50 bp of its start and end coordinates), Internal (3′ peaks located on the same strand within an annotated mRNA ORF, tRNA, rRNA or sRNA coordinates, excluding the 3′ end coordinate) and Orphan (3′ peak not falling in any of the previous classes).

3′ ends were also categorized according to their position relative to mRNA 5′ UTRs and internal mRNA regions (Supplementary Data 5). Any 3′ end (Supplementary Data 1) that was located directly upstream of an ORF (200 bp upstream of an annotated start codon) or ORF-internal was extracted and further analyzed. To remove any 3′ ends that likely belonged to an upstream gene in the same direction, TSS data[37] obtained using the same growth condition and *B. burgdorferi* strain as 3′RNA-seq was considered. All these 3′ ends were examined for the first upstream feature (either a TSS or an ORF stop codon). Any 3′ end where the first upstream feature was a stop codon was eliminated, unless there was also a TSS ≤200 bp upstream the 3′ end. Any 3′ end where a TSS was only 20 bp or less upstream was also eliminated. This resulted in the 3′ end coordinates in Supplementary Data 5.

## sRNA annotations

Small RNAs annotations were refined from previous studies of size-selection RNA-seq[33] or total RNA-seq[16]. For each predicted sRNA we algorithmically searched for 5′ and 3′ ends in our *B. burgdorferi* logarithmic-grown datasets, that overlapped or were sequenced within 75 nt from previous annotations. Using previous nomenclature for numbering sRNAs[33], we annotated possible TSS(s), 5′ processed end(s), and the most abundant 5′ and 3′ end for each sRNA (Supplementary Data 2). Additional case-by-case curation was performed as follows. tRNAs, which were misannotated as sRNAs in one study[33], were eliminated from this list. sRNA annotations that overlapped were manually examined to determine if they were in reference to the same sRNA and merged when necessary. Two sRNAs with defined ends were predicted in Arnold et al., 2016 but not Popitsch et al., 2017 and were given new sRNA numbers (SR1006 and SR1007). All refined sRNA annotations (Supplementary Data 2) were manually inspected, and final sRNA annotation coordinates were determined by the highest sequenced 5′ and 3′ end. In a few cases the pattern of sequencing reads in the total RNA-seq samples or an algorithmically uncalled 5′ or 3′ end warranted the sRNA annotation coordinate to be manually adjusted. In total, 49 sRNAs were manually refined from the algorithmic collation.

## Updated gene annotations

*B. burgdorferi* B31 gene annotations were downloaded from NCBI, RefSeq assembly accession GCF_000008685.2_ASM868v2 ([https://www.ncbi.nlm.nih.gov/assembly/GCF_000008685.2/]); as of 2020-12-18. sRNA annotations (Supplementary Data 2), as outlined above, were merged and sRNA gene names included for *ssrA*/tmRNA (SR0030), *ssrS* (6 S) (SR0136)[108], *rnpB* (SR0613), BsrW (SR0412)[16], and IttA (SR0736)[28]. The following sRNAs were missed from Supplementary Data 2 because of a lack of a 3′/5′ end within 75 nt of the previous sRNA annotations and were manually added: *srp*, *ffs* (4.5 S) (SR0360), DsrA (SR0440)[27] and SR0726 [109]. The ORF annotation for *pncA* (*bbe22*) was updated to include the experimentally determined start codon[110]. 5′ UTRs were predicted by the presence of a TSS, with the highest average TAP+ sequencing read[37], within a window of 300 nt upstream of an annotated start codon. 3′ UTRs were predicted by the presence of a primary 3′ end 100 nt downstream of an annotated stop codon. UniProt gene names and descriptions from the UniProt *B. burgdorferi* B31 database (Proteome ID: UP000001807; as of 2022-08-03) were imported to supplement the gene information reported from NCBI. Reference to gene *bb0268* as '*hfq*' was removed, as this gene was previously misannotated. Collectively, these data resulted in Supplementary Data 3.

## Analysis of 3′ ends

Each identified 3′ end and its surrounding sequence (−50 nt to +10 nt, relative to a 3′ end) was analyzed using a previously described quantitative model[48] to score putative intrinsic terminators, which uses KineFold to simulate the kinetic folding of mRNA as it is transcribed by RNAP[111]. The results for each 3′ end region are included in Supplementary Data 1. The G/C content for each sequence was determined by reporting the strand-specific percentage of GC over a sliding 5 nt-window, with a step of 1 nt. The 3′ ends were sorted according to the position of the highest GC content and displayed in a heatmap (using the heatmap function from the seaborn[112] package 0.11.). The average GC per position is given below the heatmap. Terminator logos were generated using MEME[113] 5.4. with the RNA-specific default.

As detailed in Fig. S4, we compared the *B. burgdorferi* 3′ ends identified by 3′RNA-seq with 3′ ends and predicted terminators

identified in other studies. Specifically, we determined how many of the positions in our list of genome coordinates are within a given distance threshold of a genome coordinate from another study.

To compare features of *B. burgdorferi* termination to other bacteria, we analyzed raw 3'RNA-seq (Term-seq) data from *E. coli* ([12]; LB 0.4, SRA accession number PRJNA640168), *P. aeruginosa* ([50]; Paer_PAO1_60-AHLs_term_seq replicates 1 – 3; SRA accession numbers ERR3258013-ERR3258015 [https://www.ncbi.nlm.nih.gov/bioproject/PRJEB31965]) and *B. subtilis* ([18]; lib1, lib2, lib10 and lib11 samples; SRA accession numbers ERS1048762, ERS1051962, ERS1051954, ERS1051963 [https://www.ncbi.nlm.nih.gov/bioproject/PRJEB12568]) were downloaded with sra-toolkit fastq-dump v2.9.1_1, and processed according to our established "Identification of 3' ends from Term-seq" pipeline. The published *E. coli* data (previous work from our groups[12]) was analyzed using lcdb-wf v1.7. To keep the analysis consistent between all organisms and avoid version discrepancies, the raw *E. coli* sequencing data was re-analyzed using lcdb-wf v1.7, specifically using the same versions of all software dependencies to ensure consistency across analyses. G/C content and putative terminator logos for other bacteria were analyzed the same as *B. burgdorferi* 3' end sequences.

## Total RNA-seq

Total RNA-seq was performed using the same RNA that was used for the 3'RNA-seq library preparations. Total RNA-seq library construction was carried out based on the RNAtag-seq methodology[41], which was adapted to capture bacterial sRNAs[101]. Total RNA-seq RNA libraries were sequenced as for 3'RNA-seq. Total RNA-seq data processing followed the same procedures as 3'RNA-seq data analysis for QC, adaptor removal and sequencing read mapping.

## BCM treatment and ± BCM-seq

Two cultures of *B. burgdorferi* B31 (PA003) spirochetes were grown in BSKII to a cell density of $4.0 \times 10^7$ (replicate #1) or $2.75 \times 10^7$ (replicate #2) cells/ml (logarithmic phase), the cultures divided and half treated with 100 μg/ml of bicyclomycin benzoate (MedChemExpress, Catalog #50-163-6902) for 6 h. Total RNA was isolated from the untreated and BCM-treated cultures as described above. The RNA libraries were prepared and processed according to the total RNA-seq procedure, using the RNAtag-seq methodology as described above.

## Identification of Rho regions

Sequencing reads from untreated and BCM treated samples were aligned to the GCF_000008685.2_ASM868v2 genome by following the same procedures as 3'RNA-seq data analysis for QC, adaptor removal and sequencing read mapping. The approximate Rho-dependent transcription termination sites were predicted by identifying the locations of transcriptional readthrough in the BCM treated sample using rhoterm-peaks[12] (available at https://github.com/gbaniulyte/rhoterm-peaks). The ratio of the read count in a 800 nt region upstream (BCM_us) and downstream (BCM_ds) of each position was compared to the same ratios at the same position in the untreated sample (Untreated_us, Untreated_ds) ($R_{(BCM/untreated)} = \frac{BCM\_ds/BCM\_us}{Untreated\_ds/Untreated\_us}$). The ratio $R_{(BCM/Untreated)}$ threshold for termination sites was set to ≥2.0. Only the position with the highest $R_{(BCM/Untreated)}$ value within an 800 nt window upstream and downstream was reported in Supplementary Data 4. A window size of 800 nt was selected as large enough to have a sufficient number of reads for most regions and small enough to identify one Rho-termination site per transcriptional unit. *P*-values were calculated by a two-sided Fisher exact test. An 'undefined' Rho score indicates one that could not be calculated due to an absence of reads in one or more ±BCM 800 nt upstream or downstream regions. A significance score of 'n/a' indicates that the Rho significance score was too low ($<1e^{-300}$) to accurately report and indicates a highly significant score. No significance scores were calculated for undefined scores. The presence of a Rho region

was also indicated for each gene when one was identified on the same strand internal to or 800 bp downstream of an mRNA ORF, tRNA or sRNA, in either or both RNA-seq replicates (Supplementary Data 4, tab: "Rho regions – by gene").

## Spermidine treatment and ±spermidine-seq

Three cultures of *B. burgdorferi* B31 (PA003) spirochetes were diluted into BSK-II or BSK-II + 10 mM spermidine (Sigma Aldrich, Catalog #S2626-1G) adjusted to pH 7.8. Cells were collected in stationary phase ( ~ $1 \times 10^8$ – $2 \times 10^8$ cells/ml) after ~80 h of growth in BSK-II alone or ~96 h of growth in BSK-II + 10 mM spermidine. Total RNA was isolated from the untreated and spermidine-treated cultures as described above. The RNA libraries were prepared and processed according to the total RNA-seq procedure using the RNAtag-seq methodology as described above and sequenced using the Nova-seq system.

## Differential expression in ±spermidine conditions

Sequencing reads from untreated and spermidine-treated samples were aligned to the GCF_000008685.2_ASM868v2 genome by following the same procedures as 3'RNA-seq data analysis for QC, adaptor removal and sequencing read mapping. Raw counts were quantified using featurecounts from the subread package v.2.0.1. Numerous gene features overlap other features in *B. burgdorferi*; to estimate the expression of such overlapping genes, the quantification was performed two ways, one that included reads overlapping 2 or more features (parameter "−allowMultiOverlap"), and one performed with standard quantification parameters that included only reads overlapping a single feature. Differential expression was performed (Supplementary Data 6) using raw counts provided to DESeq2 v1.30.0 with the following modifications from lfcShrink default parameters: type = "normal". False discovery rate (FDR) was applied to two-sided Wald test *p*-values generated by DESeq2 to determine significance. A gene was considered differentially expressed if the false discovery rate (FDR) was <0.1 (default for DESeq2) for the statistical test that the magnitude of the log$_2$-fold change is greater than 0 (lfcThreshold=0, default for DESeq2).

## Identification of spermidine-dependent 3' ends

Processed sequence reads from "Differential expression in ±spermidine conditions" were used. Spermidine-dependent scores were calculated as Rho region scores, with the difference that the minus spermidine ratios were compared to the plus spermidine ratios ($R_{(untreated/spermidine)} = \frac{untreated\_ds/untreated\_us}{spermidine\_ds/spermidine\_us}$). The ratio $R_{(untreated/spermidine)}$ threshold for spermidine dependency was set to ≥ 2.0. All upstream and ORF-internal 3' ends were given a spermidine-dependent score (Supplementary Data 5) by reporting the $R_{(untreated/spermidine)}$ value calculated across a 50 nt window upstream and downstream of each 3' end. *P*-values were calculated by a two-sided Fisher exact test. An 'undefined' spermidine-dependent score indicates one that could not be calculated due to an absence of reads in one or more ±spermidine 50 nt upstream or downstream regions. A significance score of 'n/a' indicates that the spermidine-dependent significance score was too low ($<1e^{-300}$) to accurately report and indicates a highly significant score. No significance scores were calculated for undefined scores.

## Northern blot analysis

Northern blots were performed using total RNA exactly as described previously[100]. For sRNAs, 5 μg of RNA were fractionated on 8% polyacrylamide urea gels containing 6 M urea (1:4 mix of Ureagel Complete to Ureagel-8 (National Diagnostics) with 0.08% ammonium persulfate) and transferred to a Zeta-Probe GT membrane (Bio-Rad). For longer RNAs, 10 μg of RNA were fractionated on a 2% NuSieve 3:1 agarose (Lonza), 1X MOPS, 2% formaldehyde gel and transferred to a Zeta-Probe GT membrane (Bio-Rad) via capillary action overnight. For both

types of blots, the RNA was crosslinked to the membranes by UV irradiation. RiboRuler High Range and Low Range RNA ladders (Thermo Fisher Scientific) were marked by UV-shadowing. Membranes were blocked in ULTRAhyb-Oligo Hybridization Buffer (Ambion) and hybridized with 5′ $^{32}$P-end labeled oligonucleotides probes (listed in Supplementary Data 7). After an overnight incubation, the membranes were rinsed with 2X SSC/0.1% SDS and 0.2X SSC/0.1% SDS prior to exposure on film. Blots were stripped by two 7-min incubations in boiling 0.2% SDS followed by two 7-min incubations in boiling water.

## Immunoblot analysis
Immunoblot analysis was performed as described previously[114] with minor changes. Samples were separated on a Mini-PROTEAN TGX 5%–20% Tris-Glycine gel (Bio-Rad) and transferred to a nitrocellulose membrane (Thermo Fisher Scientific). Membranes were blocked in 1X TBST containing 5% milk, probed with a 1:2,000 dilution of polyclonal α-PotA[65] followed by a 1:10,000 dilution of peroxidase labeled α-mouse (GE Healthcare) and developed with SuperSignal West Pico PLUS Chemiluminescent Substrate (Thermo Fisher Scientific) on a Bio-Rad ChemiDoc MP Imaging System.

## Mouse and tick infections
For the isolation of spirochete RNA from ticks, six 6-8 week old female C3H/HeN mice (Envigo) were needle inoculated intraperitoneally (80%) and subcutaneously (20%) with $1 \times 10^5$ B. burgdorferi B31 A3 (PA003) spirochetes. The inoculum dose was verified by colony-forming unit (CFU) counts in solid BSKII medium. Three weeks post inoculation, infection was confirmed by serology, as previously described[115]. Approximately 200 uninfected Ixodes scapularis larvae (Oklahoma State University) were allowed to feed to repletion on the B. burgdorferi infected mice, as previously described[57,116]. Two weeks from the tick placement, groups of 25 fed larvae were pooled, snap frozen in liquid nitrogen, and processed for RNA isolation. The remaining fed larvae were maintained at 23 °C under 98% humidity and allowed to molt to nymphs. Four groups of 25 nymphs each were fed to repletion on individual naïve 6–8 week old female C3H/HeN mice. Two weeks from the tick placement, groups of 5 fed nymphs were pooled, snap frozen in liquid nitrogen, and processed for RNA isolation. For the isolation of spirochete RNA from mouse tissues, six 6–8 week old female C3H/HeN mice (Envigo) were needle inoculated intraperitoneally (80%) and subcutaneously (20%) with $1 \times 10^5$ B. burgdorferi B31 A3 (PA003) spirochetes. Ten days post inoculation mice were sacrificed and groups of 2 mouse bladders were pooled, snap frozen in liquid nitrogen, and processed for RNA isolation. Infection was confirmed by the reisolation of spirochetes from ear, heart, and joint tissues.

## Ethics statement
The National Institutes of Health is accredited by the International Association for Assessment and Accreditation of Laboratory Animal Care. All animals were cared for in compliance with the Guide for the Care and Use of Laboratory Animals. Protocols for all animal experiments were prepared according to the guidelines of the National Institutes of Health, reviewed and approved by the *Eunice Kennedy Shriver* National Institute of Child Health and Human Development Institutional Animal Care and Use Committee.

## RT-qPCR analysis
Three biological replicates of B. burgdorferi B31 (PA003) were grown to mid-logarithmic phase ($2.75 \times 10^7$ – $5.25 \times 10^7$ cells/ml), the cultures split for RNA isolation or allowed to grow an additional 48 h to stationary phase ($2.18 \times 10^8$ – $3.08 \times 10^8$ cells/ml) in BSKII. Cells were pelleted at the indicated time points, washed once with 1X PBS, and snap frozen in liquid $N_2$. Frozen culture-pellets, tick, or mouse bladder samples were homogenized in ~1 ml TE buffer (10 mM Tris,

1 mM EDTA, pH 8.0) containing 0.5 mg/ml lysozyme in gentleMACS M Tubes using a gentleMACS Dissociator (program RNA_02; Miltenyi Biotech). RNA was isolated using a hot phenol/SDS protocol. Briefly, homogenized samples were mixed and incubated with 1% SDS, 0.1 M NaOAc, pH 5.2 and equal parts saturated phenol (pH 6.6) at 64 °C for 6 min. Samples were centrifuged for 10 min at 4 °C on maximal speed and the upper phase ( ~ 0.6 ml) was transferred into a Heavy Phase Lock Gel tube (5 PRIME) with 800 µl of buffer-saturated phenol:chloroform:isoamyl alcohol (25:24:1). Samples were mixed thoroughly by inversion and centrifuged for 7 min at 4 °C on maximal speed. The aqueous layer was transferred to a new Heavy Phase Lock Gel tube and 800 µl of chloroform was added. Samples were mixed and spun again. The aqueous layer was removed and ethanol precipitated. RNA pellets were reconstituted in DEPC $H_2O$. For tick RNA samples, any contaminating DNA was removed by treating 1–5 µg of RNA with the TURBO DNase enzyme kit and purified according to the manufacturer's instructions (ThermoFisher Scientific). For mouse bladder RNA samples, any contaminating DNA was removed by treating 25 µg of RNA with 10 U of DNase I (Roche) for 20 min and purified as previously described[12].

cDNA was synthesized from DNase-treated RNA from cell-culture, fed larvae, fed nymph and mouse bladder samples using the iScript Reverse Transcription supermix kit according to the manufacturer's instructions (Bio-Rad). Parallel cDNA reactions were carried out in the absence of reverse transcriptase. Real-time quantitative PCR reactions were performed using SsoAdvanced universal SYBR green supermix and the following primer pairs: PA630 and PA634 (potB 5′), PA635 and PA636 (potB ORF 3′), PA495 and PA496 (ospC), and PA489 and PA490 (recA). Reactions were placed in CFX Opus 96 (Bio-Rad) with the following program specifications: Step 1: 95 °C 5 min; Step 2: 95 °C 10 sec, 55 °C 30 sec – repeat Step 2 for 40 cycles; Step 3: 95 °C 15 sec, 55 °C to 95 °C melt curve. The amounts of each transcript were determined using a genomic DNA standard curve for each gene target with 10 ng, 1 ng, 0.1 ng, and 0.01 ng of DNA. Both standard-curve reactions and PCRs using cDNA for each biological replicate were performed in technical duplicate or triplicate. mRNA transcript copy numbers were normalized to recA copies. See Source Data file for raw data.

## Luciferase assays
Luciferase assays were modified from[37,117]. B. burgdorferi clones were subcultured to a density of $1 \times 10^5$ cells/ml in 15 ml of BSKII and after 48 hours of growth ($1.0 \times 10^7$ – $3.5 \times 10^7$ cells/ml), pelleted at 4000 rpm for 10 min. Cells were resuspended in 15 ml of BSKII containing either no spermidine, 5 mM spermidine or 10 mM spermidine (Sigma Aldrich, Catalog #S2626-1G). Prior to resuspension, media containing spermidine was pH adjusted to 7.8 to match BSKII not containing spermidine. After an additional ~40 h of growth ($1.1 \times 10^8$–$1.7 \times 10^8$ cells/ml), cells were pelleted at 4000 rpm for 10 minutes, washed with phosphate-buffered saline (PBS) (137 mM NaCl, 2.7 mM KCl, 10 mM $Na_2HPO_4$, 1.8 mM $KH_2PO_4$, pH 7.4) and resuspended in 225 µl of PBS. 100 µl of each sample was used to measure the optical density at 600 nm ($OD_{600}$) using an EnVision 2105 Multimode Plate Reader (PerkinElmer). 93 µl of each sample were loaded into a black, solid bottom 96-well plate (Corning) and combined with 7 µl of 10 mM D-luciferin (Regis) in PBS. The relative luciferase units (RLUs) were determined by measuring photon emission in each well for 2 s using the EnVision 2105 Multimode Plate Reader, 5 min after the addition of luciferin. The background RLU was determined by measuring a PBS control, which was subtracted from all experimental measurements. Background-subtracted RLUs were normalized to the $OD_{600}$ value of the same sample ($RLU/OD_{600}$). All experiments were conducted in biological triplicate. B. burgdorferi harboring an empty vector (PA156) was used as a control for all experiments. See Source Data file for raw data and statistical analysis.

**Reporting summary**

Further information on research design is available in the Nature Portfolio Reporting Summary linked to this article.

## Data availability

The raw sequencing data reported in this paper have been deposited in GEO under accession number GSE222088. The SuperSeries is composed of the following SubSeries:- GSE222084 Extensive diversity in RNA termination and regulation revealed by transcriptome mapping for the Lyme pathogen B. burgdorferi [bulk RNA-seq]- GSE222085 Extensive diversity in RNA termination and regulation revealed by transcriptome mapping for the Lyme pathogen B. burgdorferi [BCM RNA-seq] - GSE222086 Extensive diversity in RNA termination and regulation revealed by transcriptome mapping for the Lyme pathogen B. burgdorferi [SPD RNA-seq] - GSE222087 Extensive diversity in RNA termination and regulation revealed by transcriptome mapping for the Lyme pathogen B. burgdorferi [3′RNA-seq]. The previously published data were analyzed: 3′RNA-seq (Term-seq) data from E. coli (PRJNA640168), P. aeruginosa (ERR3258013-ERR3258015 [https://www.ncbi.nlm.nih.gov/bioproject/PRJEB31965]) and B. subtilis (ERS1048762, ERS1051962, ERS1051954, ERS1051963 [https://www.ncbi.nlm.nih.gov/bioproject/PRJEB12568]). Source data for this paper have been submitted to Figshare (https://doi.org/10.6084/m9.figshare.22569205), which includes all uncropped and unprocessed scans with molecular weight markers labeled. The processed RNA-seq data from this study are available online via UCSC genome browser at the following links: ***B. burgdorferi* 3′RNA-seq (logarithmic and TS-stationary cells):** https://www.nichd.nih.gov/about/org/dir/other-facilities/cores/bioinformatics/data/adams-borrelia-three-prime. 3 total RNA-seq tracks for logarithmic cells (top + bottom strand). 3 3′RNA-seq tracks for logarithmic cells (top + bottom strand). 3 total RNA-seq tracks for TS-stationary cells (top + bottom strand). 3 3′RNA-seq tracks for TS-stationary cells (top + bottom strand). ***B. burgdorferi* RNA end mapping (logarithmic growth cells):** https://www.nichd.nih.gov/about/org/dir/other-facilities/cores/bioinformatics/data/adams-borrelia-RNA-mapping-log 2 5′RNA-seq -TAP tracks for logarithmic cells (top + bottom strand). 2 5′RNA-seq +TAP tracks for logarithmic cells (top + bottom strand). 3 total RNA-seq tracks for logarithmic cells (top + bottom strand). 3 3′RNA-seq tracks for logarithmic cells (top + bottom strand). ***B. burgdorferi* Rho-dependent 3′ ends (logarithmic phase cells):** https://www.nichd.nih.gov/about/org/dir/other-facilities/cores/bioinformatics/data/adams-borrelia-rho-dependent. 2 -BCM-seq tracks (top + bottom strand). 2 +BCM-seq tracks (top + bottom strand). 1 3′RNA-seq track representing overlay/average (top + bottom strand). ***B. burgdorferi* ±spermidine total RNA-seq (stationary phase cells):** https://www.nichd.nih.gov/about/org/dir/other-facilities/cores/bioinformatics/data/adams-borrelia-spermidine. 3 -spermidine total RNA-seq tracks (top + bottom strand). 3 +spermidine total RNA-seq tracks (top + bottom strand).

## Code availability

All code for data processing has been previously reported and is available on Github [https://github.com/lcdb/lcdb-wf/releases/tag/v1.7].

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

## Acknowledgements
We thank M. W. Jewett, J. Wade, and members of the Adams and Storz labs for helpful comments on the manuscript. We thank J. Seshu for providing the PotA antibodies. We appreciate the support from numerous members of the NIH Division of Occupational Health and Safety, the NIH Institutional Biosafety Committee, the NICHD Animal Care and Use Committee, the NIH Division of Veterinary Resources especially R. Baumann, C. Buckwalter, R. Kastenmayer, K. Pfeifer, K. Eckard, J. Smith, M. Tenace, M. Bermudez and W. Torres Gomez for the development and execution of the tick and mouse *B. burgdorferi* infections performed in this study. We thank the NICHD Molecular Genomics Core, particularly T. Li, for library sequencing. This work utilized the computational resources of the NIH HPC Biowulf cluster (http://hpc.nih.gov). This work was supported by the National Institutes of Health Independent Research Scholar Program and the *Eunice Kennedy Shriver* National Institute of Child Health and Human Development Intramural Research Program [1ZIAHD008995-02 to P.P.A.]; a National Institute of General Medical Sciences Postdoctoral Research Associate (PRAT) fellowship [1Fi2GM133345-01 to P.P.A]; the *Eunice Kennedy Shriver* National Institute of Child Health and Human Development Intramural Research Program [1ZIAHD01608-31 to G.S.] and the Bioinformatics and Scientific Programming Core [1ZICHD008986-03 to R.K.D].

## Author contributions
P.P.A. and G.S. conceived the project. P.P.A., G.S., and R.K.D. funded the project. P.P.A. and G.S. designed the experiments. P.P.A., and E.P. performed the experiments. C.E., R.K.D., and D.T. performed the bioinformatics analysis. P.P.A., G.S., C.E., E.P., and D.T. wrote the manuscript. All authors critiqued and edited the final manuscript.

## Funding

## Competing interests
The authors declare no competing interests.
