## [Peer Review File · Nature Communications]

Extensive diversity in RNA termination and regulation revealed
by transcriptome mapping for the Lyme pathogen *Borrelia
burgdorferi*Reviewer #1 (Remarks to the Author):

This elegant study defines the structure of the transcriptome of the Lyme disease spirochete, with particular attention to transcription termination mechanisms and regulatory small RNAs. The experiments have been carefully performed, the conclusions are justified, and the manuscript is well written: I have only a few minor concerns.

Minor points:

1. A sentence or two in the Introduction (or Discussion) about ribonucleases in *B. burgdorferi* would be worthwhile; see Anaker et al., 2018 (doi: 10.1128/JB.00035-18) and Snow et al., 2020 (doi: 10.1016/j.bbrc.2020.05.201).

2. (p. 7, lines 9-13) A temperature shift in vitro is a poor mimic of the tick to vertebrate phase transition, particularly regarding the transcriptional response; see Iyer et al., 2015. Obviously, changing this experimental condition at this point is beyond the scope of this study; however, the caveat that there are significant differences between the in vitro and in vivo transcriptomes should be clearly stated in the manuscript (despite the induction of *rpoS* transcription, the pervasive use of a temperature shift in the *B. burgdorferi* gene regulation literature, and the methodological caveats in the Iyer et al., 2015 publication, including amplification of tick-derived *B. burgdorferi* RNA and use of a microarray).

Also note that the effect of a temperature shift on *rpoS* expression predates Ellis et al., 2014; see Yang et al., 2000 (doi: 10.1046/j.1365-2958.2000.02104.x).

3. (Fig. 1C) Suggest why the RNA encoding *pncA* lacks a strong 3' end after the ORF yet has a distinct signal by Northern blotting (there is a strong 3' end after SR0754).

4. (Supplementary Table 2) What is the reason that so many sRNAs were missed in the Arnold et al., 2016 study? The explanation in the Discussion is vague (p. 27, lines 13-16).

5. Did you identify the *bmpAB* terminator described by Ramamoorthy et al., 2005?

6. (p. 22, line 7) Fig. 6D is a Northern blot, not a measure of growth.

7. (Fig. 7B) Differences in cDNA levels should be analyzed using a statistical test.

8. (Supplementary Table 2) Why was a 3'/5' end within 75 nt of the previous annotations of the sRNAs *srp*, *ffs*, *DsrA*, and *IttB* missed?

Minor editorial points:

(p. 15, line 5 and p. 21, line 11) "twofold" instead of "two-fold"

(p. 18, line 9 and elsewhere) delete "agarose" (or change to "agarose gel")

(p. 18, line 21) "adhesin" instead of "adhesion"

(p. 19, line 18 and elsewhere) "polyacrylamide" instead of "acrylamide"

(p. 23, line 18) "transcription initiation" instead of "transcription-intiation"

(Fig. 3 legend) "downstream of the" instead of "downstream the"

Reviewer #2 (Remarks to the Author):

This carefully designed study by Petroni/Esnault et al reports a genome-wide identification and analysis of transcription termination sites and regulatory RNA elements in the bacterial pathogen *Borrelia burgdorferi* – the agent of vector-borne Lyme disease. RNA 3' ends were mapped in a transcriptome-wide manner with single-nucleotide resolution using deep-sequencing, followed by careful data analyses and discussion on potential regulatory implications. In particular, the authors combined various RNA-seq methodologies (RNAtag-seq, 3'RNA-seq/Term-seq) and conditions (different growth phases, spermidine treatment, bicyclomycin (BCM) treatment) to unveil conditional transcriptional termination in *B. burgdorferi* and to differentiate between intrinsic and factor-dependent termination regions. The authors provide all data in online browsers, enabling easy access to these valuable data sets.

The authors identified a wealth of ~1,000 RNA 3' ends in logarithmic and temperature-shifted stationary phase bacteria, with the majority (63%) upstream or internal to open reading frames (ORFs), suggesting extensive regulation by premature or processed transcripts. Furthermore, they refined the genome annotation of *B. burgdorferi* with unprecedented resolution by integrating previously existing 5'RNA-seq data (PMID: 27913725) and small regulatory RNA (sRNA) predictions (PMIDs: 27706236, 28056764). Expression of several putative sRNAs, upstream, internal, or downstream to ORFs, was subsequently validated by northern blotting and growth phase-dependent transcript abundances were revealed. The authors' data indicate that transcription of some of these sRNAs seems to be terminated by Rho rather than by a canonical intrinsic Rho-independent terminator structure, which is typically found for bacterial sRNAs. The identification of Rho termination regions with a certain overlap to sites of intrinsic termination agrees with recent findings in the literature and further challenges the classical concept of these two termination mechanisms as mutually exclusive processes in bacteria.

Spermidine is the only polyamine available in the midgut of the *Borrelia* tick host. While spermidine has been reported to globally affect translation fidelity in eukaryotes, its global impact on bacteria is less studied. Here the authors demonstrate that treatment with spermidine has a large impact on the *B. burgdorferi* transcriptome. By further investigation of possible premature termination events in mRNA 5' untranslated regions (UTRs) or within coding sequences (CDSs), they report differential regulation of various RNA fragments. The authors selected the *potB* 5' RNA, transcribed at the essential spermidine importer *potABCD* locus, for more detailed investigations. While the exact mechanism remains unclear, they demonstrate that this *potB* 5' fragment accumulated upon spermidine treatment and is selectively expressed in the tick-mouse infection cycle.

Overall, this carefully performed study represents an unprecedented analysis of *B. burgdorferi* RNA boundaries and reveals novel insights into potential cis- and trans-acting transcripts, possibly involved in riboregulation of this human pathogen. Although it is a bit descriptive, it represents a very valuable resource for researchers in the field and will further advance the exploration of RNA-mediated regulatory mechanisms, which are only little understood for this important human pathogen. The presented data support the authors conclusions. Specific comments to further clarify the text and results are listed below.

Specific comments:

1) The authors report multiple 5' ends for many sRNAs in *B. burgdorferi* and refer to them as possibly processed after visual inspection (Supplementary Table 2). Please specify which criteria were used for visual inspection and which nucleases could be responsible for processing. The authors could also add some discussion about the role of ribonucleases (RNases) in sRNA regulation and RNA metabolism of *B. burgdorferi*. For example, RNase III was reported to impact mRNA turnover (PMID: 32703440).

2) Please add some explanations on why the Rho termination regions and scores are so different between replicates while the PCA plot suggests strong correlation (Fig. S5B). This is especially evident after comparison of Rho region positions (Supplementary Table 4). Furthermore, it seems that all Rho termination regions were used for the calculation of local G:C ratios in Fig. 4A without considering the cut-offs for significance values and Rho scores calculated in Supplementary Table 4. Applying these criteria might help to improve the resolution, similar as described by the authors for *E. coli* (PMID: 33460557). In Supplementary Table 4, significance scores are indicated for undefined Rho scores, although this was excluded in the methods section (p. 41, lines 20-21). Please clarify.

3) Please add information on the respective growth phase of the samples in Figs. 3 and 4 as well as in the legends to have a better understanding for readers that are not familiar with *B. burgdorferi* cell concentrations. While the concentration of cells used for BCM-treatment is indicated with a range in the methods section, "4.0x10⁷" is specified in the legend for the northern blots in Fig. 3. Please clarify whether the concentrations used for +/-BCM-seq and northern blotting were the same.

4) In the results section "Growth-dependent effects on upstream and internal 3' ends", the authors show differential expression of 5' RNAs. Please check and revise all mentioned transcript lengths (although only stated as approximate numbers with a "~") and their annotations in Supplementary Tables 2 and 3. During comparison of the values in the tables to Figs. 5 & S6, some of the displayed lengths of 5' RNAs seemed to disagree with the text and SR0458 is missing in the annotation tables.

5) The TSSs in Fig. S6G suggest potB 5'334-561 and potB 5'485-561 might be independently transcribed. Please clarify why you refer to them as "likely 5' processed transcripts" (p. 20, line 13).

6) If the TSSs marked in Fig. S6G within the potB mRNA derive from functional promoters, probably the authors should be more careful about their conclusions regarding the potB translational reporter. This reporter setup might include possible transcriptional effects in addition to effects on translation and subsequent conclusions about post-transcriptional regulation would have to be more carefully evaluated. Please clarify.

7) In the methods section, the authors describe how RNA 3' ends were classified. However, different nucleotide lengths were used to specify primary peaks in the sections "Classification of 3' ends" (50 bp downstream of ORF) and "Updated gene annotations" (100 bp downstream an annotated stop codon). This should be clarified or explained in more detail. This is also relevant for the legend of Fig. S4, where those are specified as "100 bp downstream of the 3' end of an annotated gene".

8) Fig. 4B: From the section in the main text and the figure legend it remains a bit unclear how the authors determined the number of ORFs with Rho termination regions within or downstream of the CDS. Please clarify.

9) P. 7, line 20: With how many genes are the 1,333 and 944 3'ends detected in log or TS-stationary phase associated and how many genes does *Borrelia* have in total?

10) P. 20, line 25: As the mechanism of how spermidine leads to accumulation of 5' fragments, maybe the title of this section should be changed/softened. It is unclear if there is an effect on termination or maybe stabilization of 5' fragments and maybe no direct modulation of "generation of 3' ends". Same for p. 23, line 2. Also here the whole fragment is affected.

Text comments and suggestions:

-It could be helpful to add a sentence about the results on Rho-dependent termination and the overlap to possible intrinsic terminators in the abstract.

-P. 3, line 15: Please add references for studies on *B. burgdorferi* basic biological processes.

-It seems the results from the spermidine RNA-seq data in Supplementary Table 6 are not mentioned in the main text. This reference could be added on p. 21, line 7.

-Effects of spermidine treatment on transcriptional and translational fusions are reported in comparison to the untreated sample, whereas significance values were calculated between treated samples. Please phrase this more precisely in the main text.

-In the legend for Figure 7, the numbers for PpotA and PpotB reporters (PA345 and PA343, respectively) are different to the ones used in Supplementary Table 7 for these strains. Please check. It might also be helpful to specify in the figure legend that RT-qPCR analysis was performed on SR0541 (potB485-561) instead of "potB 5".

-Please add information to the methods section whether an additional DNase digestion step was

performed on the RNA used for total RNA-seq, +/-BCM-seq, +/-spermidine-seq, and northern blot analysis. This was included for 3'RNA-seq and RT-qPCR.

-P. 7, line 15: It is a bit unclear what the authors mean by 3'RNA-seq or Term-seq. Please clarify in the main text that a mixed protocol was used. Especially as the authors only refer to 3'RNA-seq afterwards.

-P. 23, line 18: no "-" in "transcription-initiation"

-P. 32, line 3: "termination" -> termination of transcription"

-P. 37, line 21: The described "region of 200 bp upstream of an annotated start codon to the stop" is a bit confusing. Try to rephrase.

-P. 38, lines 18-20: Please specify how many sRNA annotations had to be manually adjusted.

-Fig. 1A: A shorter fragment is visible on the northern blot below SR0885. Please add a comment on its possible identity and regulation during growth in the main text.

-Figs. 1, 6, S2: Using 5S rRNA as loading control for northern blots is not mentioned in the respective figure legends.

-Fig. 3E: The Rho termination region is indicated in the middle of the bbk15 ORF, although the text (p. 16, line 12) says "directly downstream the 5' UTR 3' end", which is confusing. Please clarify.

-Fig. 5B, genome coordinates: "608,000" instead of "608,00 bp"

-Fig. S4D: Legend lacks information on the growth phase selected for analysis of 3' UTR lengths (logarithmic phase).

-Fig. S6H: I think the longer potB fragment of the two displayed RNAs should be "potB 5'334-561".

-Fig. S6I: Information on the difference between the upper and lower panel of the northern blot is missing in the legend. And the blot seems to be the same as in Fig. 6D, not "Figure 6E".

-Fig. S7B: Maybe it is better to refer to the methods section instead of to "Figure 6E", as different amounts of spermidine and treatment times were used in Fig. 6E.

-Fig. S8B: It is unclear why the PotB amino acid sequence is shown in italics except for codons 4-7.

-Fig. S8C, legend: The secondary structure of only potB485-561 (SR0541)(and not 1-561) is shown. Please adjust the legend and phrase more precisely in the main text (p. 24, line 5).

-Supplementary Tables 1 and 5: Maybe introduce abbreviation "AVG" in the table legends.

-Supplementary Table 4: Maybe indicate in the legend that ""0" indicates that no Rho region was found in neither of the replicate data sets".

-Supplementary Table 6 is named "Table S6" in the excel file in contrast to all other Supplementary Tables and respective legends.

-Supplementary Table 7: The legend would benefit from more information on the abbreviations used, especially for "NRT" and whether "n/a" means no amplification at all. It might be useful to include the threshold cycle values for the potB ORF 3' RNA also for tick and mouse samples, although it was below the detection limit.

Reviewer #3 (Remarks to the Author):

The manuscript by Petroni et al. describes a series of transcriptomics experiments in the Lyme disease pathogen *Borrelia burgdorferi*. The lengthy manuscript was generally well written and enjoyable to read. The results globally identified 3' ends of transcripts via Term-seq and combined that data with previously generated 5' end mapping data to derive a transcriptome for this organism. Results also demonstrate the combined power of transcriptomics followed up with Northern analysis. The work also improved the annotation of protein coding genes and several sRNAs, some of which could be regulatory. The study also identified several 3' ends in 5'UTRS (5' leaders, see below) that suggest attenuation/antitermination mechanisms. All this information will be highly useful for anyone studying gene expression in *B. burgdorferi*.

Using the combined information along with standard RNA-seq data generated in this study, the authors sought to identify intrinsic transcription terminators based on the 3' ends that were mapped. The authors also perform similar experiments when cells were treated with bicyclomycin, an inhibitor of the Rho termination factor. This information was used to identify likely regions of Rho-dependent termination. Additional studies were performed when cultures were grown with and without spermidine to assess its effect on the transcriptome. Lastly, experiments were

conducted using an infection model to assess expression of one interesting transcript identified in the transcriptomics work (potB).

Important issues to consider

1. I don't have a lot of confidence on the method used to identify intrinsic terminators. The method generates a score value and anything above 3 was considered an intrinsic terminator. For example, the putative intrinsic terminator with a 3' end position of 182 has a high terminator score of 6.87. However, this putative intrinsic terminator has what one would consider a weak RNA hairpin ($\Delta G = -1.30$ kcal/mol) and poor U-tract (UAAAUUAA). Some of the predicted terminators with score values above 3 look reasonable, while many others do not. To be sure, almost nothing is known about intrinsic termination in this organism and the "rules" of termination may not follow what has been observed for *E. coli* or *B. subtilis*. Some effort needs to be invested into identifying what constitutes an authentic intrinsic terminator. The most substantial missing component of this study is the lack of in vitro validation for even a single predicted terminator. A paper was cited in this manuscript indicating that RNA polymerase has been purified from this organism and tested in vitro and I believe it is essential to do so for this study.

2. It is difficult to "call" sites of Rho-dependent termination and it seems to me that this manuscript does as good a job at doing that as any that I have seen.

3. I don't feel that the rationale for the spermidine studies were well described, and the results do not add much to the goal of the study, which was to investigate transcription termination.

4. Did you look for potential antiterminator structures in the 5' leaders where you identified putative intrinsic terminators?

5. It seems that many if not most of the 3' ends identified in this study could have been derived from mRNA processing/degradation. While the authors suggest the possibility of 3' to 5' exonucleases being responsible for some of their mapped 3' ends in the Discussion, the likely possibility of 3' end formation caused by endonucleases was not mentioned. Even the role of exonucleases was only mentioned in passing rather than it being given the prominence that it probably deserves.

6. While 5'UTR is often used to indicate 5' leader regions in the literature, many of these so-called UTRs are translated (uORF). This is particularly problematic here because 5'UTR and uORF is mentioned in the same sentence in the Discussion. (Page 31, line 14) ...spermidine induces ribosome stalling at a uORF in the 5'UTR...

You can't have an ORF in a UTR.

7. The display of the heat maps in Fig. 2 is beautiful. The yellow bands just pop.

Minor issues

1. The Dar et al. paper was not the first to describe the method for mapping of 3' ends, although they did coin Term-seq. Perhaps the paper that originally developed the method should be cited (PMID: 27571753). Another paper later combined 5' and 3' end mapping in a single method called Rend-seq (PMID: 29606352).

2. Page 12, line 15 and elsewhere. This might be an issue of personal preference, but I think that ...downstream each... should be ...downstream of each...

3. Page 18, line 4 and in the Discussion. How can identification of RNA 3' ends identify RNA thermometers, which are involved in regulating translation initiation?

4. Page 21, line 7 (Fig. 7B). Would a standard volcano plot be a better way to present the data?

5. Page 22, line 7. Fig. 6D is a Northern, not a growth curve. A growth curve would be a useful addition here.

6. Page 22, line 15 and elsewhere. What do you mean by a "premature RNA"? If you mean premature termination involved in regulated termination you should describe it as such.
7. Page 28, lines 8-10. I don't see how this sentence is connected to what is written above.
8. Page 30, lines 1-6. I don't understand what you mean. How does inhibition of Rho, which would cause transcription readthrough, lead to shorter transcripts?
9. Why were cells harvested when grown at 35°C rather than at 37°C?
10. Page 41, lines 19-20 and elsewhere. I don't understand what you mean. The lower the p value, the higher the significance. Here you are suggesting the opposite.
11. Figure 2 legend. A's and U's should be plural, not possessive. I suggest Us or U residues.
12. When you display data (e.g. Fig. 3) you overlap the replicates. Why not show averages of the replicates. An overlay results in only being able to see the replicate with the highest value.
13. Figure 6D, bottom labels. The exponential values won't be visible in print.
14. Fig. S4B legend. I don't understand how they fit into two categories. Perhaps a little more explanation would be helpful.
15. Fig. S6I legend. Indicate that potA is on the top and potB is on the bottom.
16. Fig. S8C legend. this structure is labeled as SR0541, not potB.

RESPONSE TO REVIEWER COMMENTS

Reviewer #1 (Remarks to the Author):

This elegant study defines the structure of the transcriptome of the Lyme disease spirochete, with particular attention to transcription termination mechanisms and regulatory small RNAs. The experiments have been carefully performed, the conclusions are justified, and the manuscript is well written: I have only a few minor concerns.

We appreciate these positive comments.

Minor points:

1. A sentence or two in the Introduction (or Discussion) about ribonucleases in *B. burgdorferi* would be worthwhile; see Anaker et al., 2018 (doi: 10.1128/JB.00035-18) and Snow et al., 2020 (doi: 10.1016/j.bbrc.2020.05.201).

A description of what is known about ribonucleases in *B. burgdorferi* has been added to the discussion (beginning on page 28; line 24).

2. (p. 7, lines 9-13) A temperature shift *in vitro* is a poor mimic of the tick to vertebrate phase transition, particularly regarding the transcriptional response; see Iyer et al., 2015. Obviously, changing this experimental condition at this point is beyond the scope of this study; however, the caveat that there are significant differences between the *in vitro* and *in vivo* transcriptomes should be clearly stated in the manuscript (despite the induction of *rpoS* transcription, the pervasive use of a temperature shift in the *B. burgdorferi* gene regulation literature, and the methodological caveats in the Iyer et al., 2015 publication, including amplification of tick-derived *B. burgdorferi* RNA and use of a microarray).

We acknowledge that temperature shifting bacterial cultures is a poor mimic of the tick to vertebrate phase transition and have added sentences to the discussion (page 24; lines 9-14) to comment on this limitation. However, as mentioned by the reviewer, temperature shifting is a culture condition used to study aspects of gene regulation in *B. burgdorferi*, and we observe differences in RNA levels between cultures grown to stationary phase and cultures temperature shifted prior to stationary phase (Fig. 5).

Also note that the effect of a temperature shift on *rpoS* expression predates Ellis et al., 2014; see Yang et al., 2000 (doi: 10.1046/j.1365-2958.2000.02104.x).

This reference was added (page 6; line 12).

3. (Fig. 1C) Suggest why the RNA encoding *pncA* lacks a strong 3' end after the ORF yet has a distinct signal by Northern blotting (there is a strong 3' end after SR0754).

As for any techniques, there are some limitations in RNA-seq (for example: biases in RNA adaptor ligation/cDNA amplification), which is why RNA-seq data do not always perfectly match northern analysis. We now comment on this in the discussion (page 25; lines 2-6).

4. (Supplementary Table 2) What is the reason that so many sRNAs were missed in the Arnold et al., 2016 study? The explanation in the Discussion is vague (p. 27, lines 13-16).

Arnold et al., 2016 performed total RNA-seq alone and missed many sRNAs derived from other transcripts (ORF-internal, 5' and 3' sRNAs). We have added this statement to the discussion of sRNA annotations (page 26; lines 1-5).

5. Did you identify the *bmpAB* terminator described by Ramamoorthy et al., 2005?

Yes, we also detected the 3' end internal to *bmpB* as reported in the Ramamoorthy et al., 2005 study. We have now referenced this in the discussion of ORF-internal 3' ends (page 26; lines 19-20)

6. (p. 22, line 7) Fig. 6D is a Northern blot, not a measure of growth.

We wanted to provide information about the growth phase of the samples analyzed by the northern blot. We have now clarified this in the text (page 20; lines 16-17) and increased the font size and changed the position of the cell density labels in Fig. 6.

7. (Fig. 7B) Differences in cDNA levels should be analyzed using a statistical test.

This analysis has now been performed and added to the figure.

8. (Supplementary Table 2) Why was a 3'/5' end within 75 nt of the previous annotations of the sRNAs *srp*, *ffs*, *DsrA*, and *ltdB* missed?

It is likely ends will be missed on the basis of the thresholds chosen for peak detection and the RNA-seq filtering algorithm. The sRNAs mentioned above were missed for various reasons. The sRNA *srp* (*ffs*) was missed because of the distance threshold between 3'RNA-seq peaks. Only the 3' end peak with the highest signal is selected when peaks are within 100 bp of each other, to avoid multiple calling of noisy peak regions. *srp*'s proximity to tRNA-Arg and tRNA-Ser resulted in this sRNA being missed. In the case of *DsrA*, no TAP+ 5'RNA-seq signal was detected within the 75 bp threshold, although a 3' end was detected. We required the detection of both a 5' and 3' end to call an sRNA. A low signal was detected upstream of *ltdB* that could be interpreted as a 5' end, which was too low to be called with statistical significance.

Minor editorial points:

(p. 15, line 5 and p. 21, line 11) "twofold" instead of "two-fold"

(p. 18, line 9 and elsewhere) delete "agarose" (or change to "agarose gel")

(p. 18, line 21) "adhesin" instead of "adhesion"

(p. 19, line 18 and elsewhere) "polyacrylamide" instead of "acrylamide"

(p. 23, line 18) "transcription initiation" instead of "transcription-initiation"

(Fig. 3 legend) "downstream of the" instead of "downstream the"

Thank you for these editorial suggestions, they have all been corrected.

Reviewer #2 (Remarks to the Author):

This carefully designed study by Petroni/Esnault et al reports a genome-wide identification and analysis of transcription termination sites and regulatory RNA elements in the bacterial pathogen *Borrelia burgdorferi* – the agent of vector-borne Lyme disease. RNA 3' ends were mapped in a transcriptome-wide manner with single-nucleotide resolution using deep-sequencing, followed by careful data analyses and discussion on potential regulatory implications. In particular, the authors combined various RNA-seq methodologies (RNAtag-seq, 3'RNA-seq/Term-seq) and conditions (different growth phases, spermidine treatment, bicyclomycin (BCM) treatment) to unveil conditional transcriptional termination in *B. burgdorferi* and to differentiate between intrinsic and factor-dependent termination regions. The authors provide all data in online browsers, enabling easy access to these valuable data sets. The authors identified a wealth of ~1,000 RNA 3' ends in logarithmic and temperature-shifted stationary phase bacteria, with the majority (63%) upstream or internal to open reading frames (ORFs), suggesting extensive regulation by premature or processed transcripts. Furthermore, they refined the genome annotation of *B. burgdorferi* with unprecedented resolution by integrating previously existing 5'RNA-seq data (PMID: 27913725) and small regulatory RNA (sRNA) predictions (PMIDs: 27706236, 28056764). Expression of several putative sRNAs, upstream, internal, or downstream to ORFs, was subsequently validated by northern blotting and growth phase-dependent transcript abundances were revealed. The authors' data indicate that transcription of some of these sRNAs seems to be terminated by Rho rather than by a canonical intrinsic Rho-independent terminator structure, which is typically found for bacterial sRNAs. The identification of Rho termination regions with a certain overlap to sites of intrinsic termination agrees with recent findings in the literature and further challenges the classical concept of these two termination mechanisms as mutually exclusive processes in bacteria. Spermidine is the only polyamine available in the midgut of the *Borrelia* tick host. While spermidine has been reported to globally affect translation fidelity in eukaryotes, its global impact on bacteria is less studied. Here the authors demonstrate that treatment with spermidine has a large impact on the *B. burgdorferi* transcriptome. By further investigation of possible premature termination events in mRNA 5' untranslated regions (UTRs) or within coding sequences (CDSs), they report differential regulation of various RNA fragments. The authors selected the *potB* 5' RNA, transcribed at the essential spermidine importer *potABCD* locus, for more detailed investigations. While the exact mechanism remains unclear, they demonstrate that this *potB* 5' fragment accumulated upon spermidine treatment and is selectively expressed in the tick-mouse infection cycle.

Overall, this carefully performed study represents an unprecedented analysis of *B. burgdorferi* RNA boundaries and reveals novel insights into potential cis- and trans-acting transcripts, possibly involved in riboregulation of this human pathogen. Although it is a bit descriptive, it represents a very valuable resource for researchers in the field and will further advance the exploration of RNA-mediated regulatory mechanisms, which are only little understood for this important human pathogen. The presented data support the authors conclusions. Specific comments to further clarify the text and results are listed below.

We appreciate this positive feedback and this reviewer's extremely careful review of our work.

Specific comments:

1) The authors report multiple 5' ends for many sRNAs in *B. burgdorferi* and refer to them as possibly processed after visual inspection (Supplementary Table 2). Please specify which criteria were used for visual inspection and which nucleases could be responsible for

processing. The authors could also add some discussion about the role of ribonucleases (RNases) in sRNA regulation and RNA metabolism of *B. burgdorferi*. For example, RNase III was reported to impact mRNA turnover (PMID: 32703440).

Supplementary Table 2 reports possible TSS(s), 5' processed end(s), and the most abundant 5' and 3' end for each sRNA. The ratio of sequencing reads in TAP+/- samples were used to distinguish an sRNA TSS from an sRNA processed 5' end, as described in our previous publication (PMID: 27913725). Given that sRNAs can exist in multiple forms and our interest in these regulators, we also manually checked the algorithmic collation of sRNA boundaries and, in 49 instances, included additional sRNA ends that were missed to more confidently report sRNAs.

A description of the ribonucleases in *B. burgdorferi* has been added to the discussion (beginning on page 28; line 24).

2) Please add some explanations on why the Rho termination regions and scores are so different between replicates while the PCA plot suggests strong correlation (Fig. S5B). This is especially evident after comparison of Rho region positions (Supplementary Table 4). Furthermore, it seems that all Rho termination regions were used for the calculation of local G:C ratios in Fig. 4A without considering the cut-offs for significance values and Rho scores calculated in Supplementary Table 4. Applying these criteria might help to improve the resolution, similar as described by the authors for *E. coli* (PMID: 33460557). In Supplementary Table 4, significance scores are indicated for undefined Rho scores, although this was excluded in the methods section (p. 41, lines 20-21). Please clarify.

PCA plots of \pm BCM-seq and the determination of Rho scores are different analyses, measuring different data. PCA plots measure the differences in total RNA reads coming from a gene, displayed in multi-dimensional space with one dimension per gene, reduced to the two most explanatory dimensions. Rho scores are calculated by sliding windows within a single sample to find peaks of reads corresponding to transcriptional readthrough (Rho regions). We do not necessarily expect the position of Rho regions (measured by Rho scores) to be closely correlated with total expression over the entire gene body (PCA). It could be that different Rho termination positions for a gene could result in largely the same total count of reads sequenced from the gene, or that Rho termination is inherently more variable than total RNA. It could also be that the algorithm for estimating Rho region positions is more sensitive to small technical variation than bulk RNA-seq data.

Regarding GC content and Rho termination regions, a Rho region only was included in the plot if it meets the significance cut-offs, which is outlined in our previous publication (PMID: 33460557). We have clarified the text on page 15, lines 13-14 to explain this.

p-values for undefined Rho scores were left in Supplementary Table 4 by mistake and have now been removed.

3) Please add information on the respective growth phase of the samples in Figs. 3 and 4 as well as in the legends to have a better understanding for readers that are not familiar with *B. burgdorferi* cell concentrations. While the concentration of cells used for BCM-treatment is indicated with a range in the methods section, " 4.0×10^7 " is specified in the legend for the northern blots in Fig. 3. Please clarify whether the concentrations used for \pm BCM-seq and northern blotting were the same.

All BCM experiments were conducted with logarithmic phase samples. We have now defined the cell density used for \pm BCM-seq in the Methods (page 38, line 18) and the cell density used in Fig. 3 in the legend. The northern blot images depicted in Fig. 3 and Fig. 4 are the same membrane, stripped and re-probed for different RNAs.

4) In the results section “Growth-dependent effects on upstream and internal 3’ ends”, the authors show differential expression of 5’ RNAs. Please check and revise all mentioned transcript lengths (although only stated as approximate numbers with a “~”) and their annotations in Supplementary Tables 2 and 3. During comparison of the values in the tables to Figs. 5 & S6, some of the displayed lengths of 5’ RNAs seemed to disagree with the text and SR0458 is missing in the annotation tables.

We checked the sizes for all sRNAs in Fig. 5 & S6. sRNAs can exist in multiple forms of different lengths, as our northern and RNA-seq analysis for these sRNAs also document. We highlighted the sequence for the most dominant sRNA product for each sRNA in Fig. S6 which was also reflected in the text by approximate numbers (“~”). This has been further clarified in the figure legend. Note that we had made an error in the labeling the predicted coding sequence of *bbk15* and the size of SR0821 – which has now been corrected (Fig. S6A). We have also clarified the labeling and description of the 5’ *bgp* sRNAs. “We detected a novel short 5’ *bgp* RNA (~58 nt) that corresponded to 5’- and 3’RNA-seq signals (Fig. 5B and S6B). This also overlapped a previously predicted longer sRNA (SR0458) of ~333 nt that may correspond to the entire *bgp* 5’ UTR (195 nt), although the boundaries for this longer sRNA are not clearly defined in our data.” (page 17; lines 11-15).

5) The TSSs in Fig. S6G suggest *potB* 5’₃₃₄₋₅₆₁ and *potB* 5’₄₈₅₋₅₆₁ might be independently transcribed. Please clarify why you refer to them as “likely 5’ processed transcripts” (p. 20, line 13).

Based on the ratio of TAP+/- sequencing reads, the 5’ ends identified for *potB* 5’₃₃₄₋₅₆₁ and *potB* 5’₄₈₅₋₅₆₁ are predicted to be processed. We have clarified this in Fig. S6G and the corresponding figure legend.

6) If the TSSs marked in Fig. S6G within the *potB* mRNA derive from functional promoters, probably the authors should be more careful about their conclusions regarding the *potB* translational reporter. This reporter setup might include possible transcriptional effects in addition to effects on translation and subsequent conclusions about post-transcriptional regulation would have to be more carefully evaluated. Please clarify.

As mentioned above, these 5’ ends are processed and were not labeled in the figure. To our knowledge there are no internal promoters in our reporter sequence.

7) In the methods section, the authors describe how RNA 3’ ends were classified. However, different nucleotide lengths were used to specify primary peaks in the sections “Classification of 3’ ends” (50 bp downstream of ORF) and “Updated gene annotations” (100 bp downstream an annotated stop codon). This should be clarified or explained in more detail. This is also relevant for the legend of Fig. S4, where those are specified as “100 bp downstream of the 3’ end of an annotated gene”.

We think this lack of clarity was due to a typo in the Methods, which has been corrected (page 35; lines 9-10).

8) Fig. 4B: From the section in the main text and the figure legend it remains a bit unclear how the authors determined the number of ORFs with Rho termination regions within or downstream of the CDS. Please clarify.

This has been clarified in the text “We observed 80 (16.2%) of Rho-termination events that overlapped with predicted intrinsic terminations (searching within and 800 bp downstream of annotated genes)” (page 15; line 19), and the figure legend.

9) P. 7, line 20: With how many genes are the 1,333 and 944 3'ends detected in log or TS-stationary phase associated and how many genes does *Borrelia* have in total?

There are a total of 1862 annotated genes. 937 genes were associated with 1204 logarithmic phase primary or internal 3' ends, and 739 gene were associated with 790 TS-stationary phase primary or internal 3' ends. Note that a 3' end can be primary to one gene and internal to another. The distributions of 3' ends are outlined in the text (page 10, lines 4-9). Total numbers of genes are listed in Supplementary Table 3.

10) P. 20, line 25: As the mechanism of how spermidine leads to accumulation of 5' fragments, maybe the title of this section should be changed/softened. It is unclear if there is an effect on termination or maybe stabilization of 5' fragments and maybe no direct modulation of “generation of 3' ends”. Same for p. 23, line 2. Also here the whole fragment is affected.

As suggested, we have changed/softened the titles of these sections to “Spermidine affects levels of ORF-internal 3' ends” (page 19) and “Spermidine-dependent regulation of *potA/B* and an ORF-internal RNA” (page 21). The mechanism by which spermidine exerts the observe effects on the transcriptome deserves further study, even with other bacteria. We performed an additional experiment to optimize the *potB* coding sequence within the translational luciferase fusion (Fig. S8B and S8D). This also disrupts the RNA secondary structure of the fusion. We no longer observed a spermidine-dependent effect with the codon optimized luciferase fusion (Fig. S8D). We have combined figures 6 and 7 to focus on *potB*, which we have examined in the most detail, and moved the panels on *bb0213* and *cdd* to supplemental Fig. S7C-E.

Text comments and suggestions:

-It could be helpful to add a sentence about the results on Rho-dependent termination and the overlap to possible intrinsic terminators in the abstract.

Thank you for the suggestion. As the overlap of Rho and intrinsic termination is just an observation and was not experimentally tested, we have elected to not include this in the abstract.

-P. 3, line 15: Please add references for studies on *B. burgdorferi* basic biological processes.

Citations to two review articles have been added.

-It seems the results from the spermidine RNA-seq data in Supplementary Table 6 are not mentioned in the main text. This reference could be added on p. 21, line 7.

Thank you for pointing this out. We have added the reference to Supplementary Table 6 as suggested.

-Effects of spermidine treatment on transcriptional and translational fusions are reported in comparison to the untreated sample, whereas significance values were calculated between treated samples. Please phrase this more precisely in the main text.

Significance values were calculated for all sample groups and are presented in Supplementary Table 7. We have added the values to the Fig. 6 comparing the treated and untreated samples.

-In the legend for Figure 7, the numbers for PpotA and PpotB reporters (PA345 and PA343, respectively) are different to the ones used in Supplementary Table 7 for these strains. Please check. It might also be helpful to specify in the figure legend that RT-qPCR analysis was performed on SR0541 (potB485-561) instead of “potB 5”.

The strain numbers in the figure legend were swapped for P_{potA} and P_{potB} reporters, which we have now corrected.

While the primers for the RT-qPCR analysis are within SR0541 they also would amplify other *potB* 5' RNAs. Therefore, we think the current label is the most accurate.

-Please add information to the methods section whether an additional DNase digestion step was performed on the RNA used for total RNA-seq, +/-BCM-seq, +/-spermidine-seq, and northern blot analysis. This was included for 3'RNA-seq and RT-qPCR.

DNase digestion is embedded as a step during library preparation for the RNAtag-seq methodology (PMID: 25730492) as detailed (PMID: 29215635), which we cite in the Methods for total RNA-seq, +/-BCM-seq, and +/-spermidine-seq. 3'RNA-seq protocols differ from RNAtag-seq in the initial steps, which is why we describe how to perform the DNase digestion step. No DNase digestion is performed on RNA prior to northern analysis.

-P. 7, line 15: It is a bit unclear what the authors mean by 3'RNA-seq or Term-seq. Please clarify in the main text that a mixed protocol was used. Especially as the authors only refer to 3'RNA-seq afterwards.

Term-seq and 3'RNA-seq refer to the same protocol. We have clarified this in the text (page 6; line 15).

-P. 23, line 18: no “-“ in “transcription-initiation“

corrected

-P. 32, line 3: “termination” -> termination of transcription”

corrected

-P. 37, line 21: The described “region of 200 bp upstream of an annotated start codon to the stop” is a bit confusing. Try to rephrase.

This has now been rephrased: “Any 3' end (Supplementary Table 1) that was located directly upstream of an ORF (200 bp upstream of an annotated start codon) or ORF-internal was extracted and further analyzed”.

-P. 38, lines 18-20: Please specify how many sRNA annotations had to be manually adjusted.

This has been specified (page 36; line 16).

-Fig. 1A: A shorter fragment is visible on the northern blot below SR0885. Please add a comment on its possible identity and regulation during growth in the main text.

We have acknowledged this band in the Fig. 1 legend and postulate it is a shorter fragment of the SR0885.

-Figs. 1, 6, S2: Using 5S rRNA as loading control for northern blots is not mentioned in the respective figure legends.

A statement about the 5S rRNA loading control has been added to these figure legends.

-Fig. 3E: The Rho termination region is indicated in the middle of the *bbk15* ORF, although the text (p. 16, line 12) says “directly downstream the 5’ UTR 3’ end”, which is confusing. Please clarify.

This has been rephrased: “a Rho termination region within the *bbk15* ORF” (page 14; lines 25-26).

-Fig. 5B, genome coordinates: “608,000” instead of “608,00 bp”

Thank you for catching this typo, we have corrected the figure.

-Fig. S4D: Legend lacks information on the growth phase selected for analysis of 3’ UTR lengths (logarithmic phase).

The growth condition is now given in the legend.

-Fig. S6H: I think the longer *potB* fragment of the two displayed RNAs should be “*potB* 5’334-561”.

Yes, this has been corrected.

-Fig. S6I: Information on the difference between the upper and lower panel of the northern blot is missing in the legend. And the blot seems to be the same as in Fig. 6D, not “Figure 6E”.

We have defined the panels in the figure and figure legend and corrected the reference to Fig. 6.

-Fig. S7B: Maybe it is better to refer to the methods section instead of to “Figure 6E”, as different amounts of spermidine and treatment times were used in Fig. 6E.

We meant to refer to Fig. 6D here (now Fig. 6B). This has been corrected in the figure legend and stated, “as performed in Fig. 6B and described in the Methods”.

-Fig. S8B: It is unclear why the *PotB* amino acid sequence is shown in italics except for codons 4-7.

This was a mistake and has been corrected.

-Fig. S8C, legend: The secondary structure of only potB485-561 (SR0541) (and not 1-561) is shown. Please adjust the legend and phrase more precisely in the main text (p. 24, line 5).

This has been adjusted in the figure legend and clarified in the main text.

-Supplementary Tables 1 and 5: Maybe introduce abbreviation "AVG" in the table legends.

We have removed this abbreviation from the tables.

-Supplementary Table 4: Maybe indicate in the legend that ""0" indicates that no Rho region was found in neither of the replicate data sets".

This is now indicated in the table legend.

-Supplementary Table 6 is named "Table S6" in the excel file in contrast to all other Supplementary Tables and respective legends.

We have corrected this label.

-Supplementary Table 7: The legend would benefit from more information on the abbreviations used, especially for "NRT" and whether "n/a" means no amplification at all. It might be useful to include the threshold cycle values for the potB ORF 3' RNA also for tick and mouse samples, although it was below the detection limit.

We have defined abbreviations below the raw data in the table. We have added threshold cycle values for the RT-qPCR to the methods.

Reviewer #3 (Remarks to the Author):

The manuscript by Petroni et al. describes a series of transcriptomics experiments in the Lyme disease pathogen *Borrelia burgdorferi*. The lengthy manuscript was generally well written and enjoyable to read. The results globally identified 3' ends of transcripts via Term-seq and combined that data with previously generated 5' end mapping data to derive a transcriptome for this organism. Results also demonstrate the combined power of transcriptomics followed up with Northern analysis. The work also improved the annotation of protein coding genes and several sRNAs, some of which could be regulatory. The study also identified several 3' ends in 5'UTRS (5' leaders, see below) that suggest attenuation/antitermination mechanisms. All this information will be highly useful for anyone studying gene expression in *B. burgdorferi*. Using the combined information along with standard RNA-seq data generated in this study, the authors sought to identify intrinsic transcription terminators based on the 3' ends that were mapped. The authors also perform similar experiments when cells were treated with bicyclomycin, an inhibitor of the Rho termination factor. This information was used to identify likely regions of Rho-dependent termination. Additional studies were performed when cultures were grown with and without spermidine to assess its effect on the transcriptome. Lastly, experiments were conducted using an infection model to assess expression of one interesting transcript identified in the transcriptomics work (potB).

We thank the reviewer for this positive summary.

Important issues to consider

1. I don't have a lot of confidence on the method used to identify intrinsic terminators. The method generates a score value and anything above 3 was considered an intrinsic terminator. For example, the putative intrinsic terminator with a 3' end position of 182 has a high terminator score of 6.87. However, this putative intrinsic terminator has what one would consider a weak RNA hairpin ($\Delta G = -1.30$ kcal/mol) and poor U-tract (UAAAUUAA). Some of the predicted terminators with score values above 3 look reasonable, while many others do not. To be sure, almost nothing is known about intrinsic termination in this organism and the "rules" of termination may not follow what has been observed for *E. coli* or *B. subtilis*. Some effort needs to be invested into identifying what constitutes an authentic intrinsic terminator. The most substantial missing component of this study is the lack of *in vitro* validation for even a single predicted terminator. A paper was cited in this manuscript indicating that RNA polymerase has been purified from this organism and tested *in vitro* and I believe it is essential to do so for this study.

Our study is the first to comprehensively identify possible intrinsic terminators in *B. burgdorferi*. We agree that the "rules" of termination in this pathogen have yet to be defined, and it is possible other protein factors are involved. We recognize the limits of using the quantitative model of predicting terminators and that future work is required as now further discussed (page 27, lines 15-18, 24-26). *In vitro* experiments to examine *B. burgdorferi* intrinsic terminators would be very valuable, however, we appreciate that the editor agrees that establishing a robust *in vitro* transcription assay for *B. burgdorferi* terminators is outside the scope of the current work.

2. It is difficult to "call" sites of Rho-dependent termination and it seems to me that this manuscript does as good a job at doing that as any that I have seen.

We agree that it is challenging to define sites of Rho termination and thank the reviewer for their comment.

3. I don't feel that the rationale for the spermidine studies were well described, and the results do not add much to the goal of the study, which was to investigate transcription termination.

The goal of our study was to (1) identify possible sites of termination and (2) to use these data to find novel RNA regulators. As reviewer #2 points out, the global impacts of spermidine in bacteria are undefined and here we show significant transcriptome consequences of excess spermidine. We have combined figures 6 and 7 to focus on *potB*, which we have examined in the most detail.

4. Did you look for potential antiterminator structures in the 5' leaders where you identified putative intrinsic terminators?

Yes, we have done this analysis and found potential anti-terminators for future study.

5. It seems that many if not most of the 3' ends identified in this study could have been derived from mRNA processing/degradation. While the authors suggest the possibility of 3' to 5' exonucleases being responsible for some of their mapped 3' ends in the Discussion, the likely possibility of 3' end formation caused by endonucleases was not mentioned. Even the role of

exonucleases was only mentioned in passing rather than it being given the prominence that it probably deserves.

A summary of ribonucleases and their contributions to RNA processing in *B. burgdorferi* has been added to the discussion (beginning on page 28; line 24).

6. While 5'UTR is often used to indicate 5' leader regions in the literature, many of these so-called UTRs are translated (uORF). This is particularly problematic here because 5'UTR and uORF is mentioned in the same sentence in the Discussion. (Page 31, line 14) ...spermidine induces ribosome stalling at a uORF in the 5'UTR...
You can't have an ORF in a UTR.

Thank you for this clarification. We have changed the language to refer to a 'leader sequence' when discussing uORFs.

7. The display of the heat maps in Fig. 2 is beautiful. The yellow bands just pop.

We were happy to read that the reviewer likes the display!

Minor issues

1. The Dar et al. paper was not the first to describe the method for mapping of 3' ends, although they did coin Term-seq. Perhaps the paper that originally developed the method should be cited (PMID: 27571753). Another paper later combined 5' and 3' end mapping in a single method called Rend-seq (PMID: 29606352).

These citations have been added.

2. Page 12, line 15 and elsewhere. This might be an issue of personal preference, but I think that ...downstream each... should be ...downstream of each...

We have made this correction as suggested.

3. Page 18, line 4 and in the Discussion. How can identification of RNA 3' ends identify RNA thermometers, which are involved in regulating translation initiation?

We previously documented, in the study cited (PMID: 33460557) in this sentence, that 3' ends have been detected even when regulation has only been reported to be at the level of translation, such as the *E. coli* RNA thermometer upstream of *rpoH*.

4. Page 21, line 7 (Fig. 7B). Would a standard volcano plot be a better way to present the data?

A volcano plot duplicates significance on both the y-axis as well as by color. In general, we prefer MA plots because they provide more information than volcano plots. Specifically, in addition to the information shown in volcano plots, MA plots also show the average number of counts, which is related to confidence in the values – and, loosely, expression level.

5. Page 22, line 7. Fig. 6D is a Northern, not a growth curve. A growth curve would be a useful addition here.

We wanted to provide information about the growth phase of the samples analyzed by the northern blot. We have now clarified this in the text and increased the font size and changed the position of the cell density labels in Fig. 6.

6. Page 22, line 15 and elsewhere. What do you mean by a "premature RNA"? If you mean premature termination involved in regulated termination you should describe it as such.

We define "premature" as an mRNA that is incomplete for translation. This has been defined on page 7, lines 23-24. The exact mechanism for generating this incomplete transcript is different for individual RNAs. In some cases, we hypothesize the incomplete transcript results from regulated termination, which we have stated when appropriate.

7. Page 28, lines 8-10. I don't see how this sentence is connected to what is written above.

We have tried to clarify our logic in this section (page 26; lines 21-24).

8. Page 30, lines 1-6. I don't understand what you mean. How does inhibition of Rho, which would cause transcription readthrough, lead to shorter transcripts?

We hypothesize that inhibition of Rho could cause increased transcriptional readthrough, generating an RNA that is targeted for degradation/processing. Shorter RNAs could be a byproduct of mRNA turnover. We have expanded the description of our hypothesis (page 28; lines 17-19).

9. Why were cells harvested when grown at 35°C rather than at 37°C?

B. burgdorferi cell are standardly cultured at 35°C.

10. Page 41, lines 19-20 and elsewhere. I don't understand what you mean. The lower the p value, the higher the significance. Here you are suggesting the opposite.

A significance score of 'n/a' was given when the Rho significance score was too low ($< 1e-300$) to accurately report and indicates a highly significant score. This has been clarified in the text (page 39; lines 13-14), (page 41; lines 4-5) and supplemental legends.

11. Figure 2 legend. A's and U's should be plural, not possessive. I suggest Us or U residues.

Thank you for pointing out this error, which has been corrected.

12. When you display data (e.g. Fig. 3) you overlap the replicates. Why not show averages of the replicates. An overlay results in only being able to see the replicate with the highest value.

Displaying the average would lose the information about variation between replicates. The overlaid replicates are distinguished by transparency. We think this will be more visible in the final high-resolution figures.

13. Figure 6D, bottom labels. The exponential values won't be visible in print.

The labels have been moved and enlarged.

14. Fig. S4B legend. I don't understand how they fit into two categories. Perhaps a little more explanation would be helpful.

Based on the organization of genes it can be ambiguous to clearly assign categories. For example, a 3' end could be directly downstream of one gene but also ORF-internal to another gene in proximity. As suggested, we have provided more explanation to this legend.

15. Fig. S6I legend. Indicate that potA is on the top and potB is on the bottom.

This has been clarified in the legend.

16. Fig. S8C legend. this structure is labeled as SR0541, not potB.

This has been corrected.

Reviewer #2 (Remarks to the Author):

The authors have carefully revised their manuscript and addressed the reviewers' comments. I have no further comments.